# Lorenza: Enhancing Generalization in Low-Rank Gradient LLM Training via Efficient Zeroth-Order Adaptive SAM

**Yehonathan Refael**                                                       *refaelkalim@tau.ac.il*
*Faculty of Engineering*
*Tel Aviv University*
*Tel Aviv 6997801, Israel.*

**Iftach Arbel**                                                              *i.arbel84@gmail.com*
*Independent Researcher*

**Ofir Lindenbaum**                                                    *ofir.lindenbaum@biu.ac.il*
*Faculty of Engineering*
*Bar-Ilan University*
*Ramat-Gan 5290002, Israel.*

**Tom Tirer**                                                              *tirer.tom@biu.ac.il*
*Faculty of Engineering*
*Bar-Ilan University*
*Ramat-Gan 5290002, Israel.*

**Reviewed on OpenReview:** *https://openreview.net/forum?id=YyA51ekcQo*

## Abstract

Modern applications often require fine-tuning large language models (LLMs) within strict memory and computational limits, but existing memory-efficient optimizers tend to compromise robustness and generalization. To tackle this, we introduce Lorenza, a low-memory optimizer based on Sharpness-Aware Minimization (SAM). Lorenza employs a stochastic zeroth-order estimator to approximate ascent directions, reducing the computational complexity of SAM while, as we prove, maintaining its convergence guarantees. Additionally, by applying randomized singular value decomposition, Lorenza performs efficient low-rank gradient updates, achieving memory efficiency similar to traditional methods. Our theoretical analysis and experiments demonstrate that Lorenza improves robustness and generalization, particularly in challenging language tasks. Furthermore, we present Lorenza+, which enhances Lorenza by incorporating the discarded orthogonal gradient component, resulting in additional performance gains without requiring extra memory or computational overhead.

## 1 Introduction

Large language models (LLMs) have garnered considerable attention due to their remarkable ability to perform various tasks, including engaging in dialogue and generating text. Their performance can be improved through supervised fine-tuning and additional pre-training across different tasks and domains. However, training these models presents substantial computational power and memory challenges. This difficulty arises because the process of updating gradients requires storing billions of trainable parameters along with the optimizer's state (which includes gradients and moments). For instance, in the Adam optimizer (Kingma & Ba, 2017), the storage requirements for gradients and the estimated first and second moments can triple the overall size of the model (Xu et al., 2024; Brown et al., 2022; Kim et al., 2023).

Researchers have developed various optimization techniques to reduce memory usage during training, tackling the challenges associated with LLM fine-tuning. One key research topic that has emerged is Parameter-Efficient Fine-Tuning (PEFT) (Han et al., 2024), which enables the adaptation of pre-trained language

models to different tasks without the need to fine-tune all model parameters. A prominent method for PEFT is the Low-Rank Adaptation (LoRA) algorithm, introduced by (Hu et al., 2021). LoRA reparameterizes a weight matrix $\mathbf{W} \in \mathbb{R}^{m \times n}$ into $\mathbf{W} = \mathbf{W}_0 + \mathbf{BA}$, where $\mathbf{W}_0$ is a frozen full-rank matrix, and $\mathbf{B} \in \mathbb{R}^{m \times r}$ and $\mathbf{A} \in \mathbb{R}^{r \times n}$ are low-rank adapters. Since $r \ll \min(m, n)$, the low-rank adapters $\mathbf{A}$ and $\mathbf{B}$ require fewer trainable parameters, reducing memory usage. Several LoRA variants have been proposed, for example (Chen et al., 2023; Xu et al., 2023; Wang et al., 2023).

Table 1: Comparison between Lorenza, GaLore (Zhao et al., 2024), LoRA (Hu et al., 2021), SAM (Foret et al., 2021), and AdaSAM (Sun et al., 2023). Assume $\mathbf{W} \in \mathbb{R}^{n \times m} (n \geq m)$, constant low-rank $r$.

|  | Lorenza | GaLore | LoRA (using Adam) | SAM | AdaSAM |
|---|---|---|---|---|---|
| Weights | $nm$ | $nm$ | $nm + nr + mr$ | $nm$ | $nm$ |
| Optimizer states | $nr + 2mr$ | $nr + 2mr$ | $2nr + 2mr$ | $nm$ | $4nm$ |
| Computation | $O\left(mnr + mnr/T\right)$ | $O\left(mnr + m^2 n/T\right)$ | $O\left(mnr\right)$ | $O\left(mnr\right)$ | $O\left(mnr\right)$ |
| Fine-Tuning | ✓ | ✓ | ✓ | ✓ | ✓ |
| Pre-Training | ✓ | ✓ | ✗ | ✓ | ✓ |
| Multi-Subspace | ✓ | ✓ | ✗ | ✗ | ✗ |
| Num. of Backprop per step | 1 | 1 | 1 | 2 | 2 |
| Sharpness-Aware | ✓ | ✗ | ✗ | ✓ | ✓ |

Despite the reduction in memory demands, low-rank weight adapters have drawbacks. One notable weakness of LoRA-type methods is their potential to fall short in accuracy for more challenging fine-tuning tasks compared to Full Fine-Tuning (FFT) (Meng et al., 2024). These issues may stem from the fact that optimal weight matrices are not inherently low-rank or from changes in gradient training dynamics introduced by the reparameterization. Another limitation, for instance, is that these methods are primarily designed for adapting pretrained LLMs through fine-tuning, rather than for use during the pretraining phase.

In an effort to mitigate these gaps, recently, efficient-memory optimizers have emerged. For example, GaLore (Zhao et al., 2024) can perform both pretraining and fine-tuning LLMs by updating the model weights within an adaptive low-rank subspace. This method significantly reduces the number of fine-tuned parameters, offering several key advantages. First, it eliminates the need for additional adapters alongside the pre-trained model. Second, it removes the requirement to store all gradient parameters during training. Third, it reduces memory usage by bypassing the need to retain the *full* dimension original optimizer states. Demonstrating its effectiveness, GaLore successfully pre-trained an LLM with 7 billion parameters on a consumer-level GPU, using just 24 GB of memory. The method attracted considerable attention, leading to various new optimizers aimed at further reducing memory usage (Refael et al., 2025; Liao et al., 2024; Zhang et al., 2024b; Das, 2024; Huang et al., 2024).

Still, a notable limitation of memory-efficient optimizers is their reduced generalization in out-of-distribution, domain shift, and zero-shot settings. This performance disparity tends to emerge in scenarios involving complex target tasks, such as mathematical reasoning or coding. This raises an open question: Can we design an optimizer that simply retains the low memory footprint of gradient descent while achieving the generalization performance of FFT-based methods?

To address this question, we introduce Lorenza, a computationally efficient, sharpness-aware optimization method that leverages adaptive low-rank gradient updates and memory-efficient zeroth-order sharpness minimization to enhance generalization, specified in Table 1. Unlike existing sharpness-aware fine-tuning methods that require costly double backpropagation (Foret et al., 2021; Sun et al., 2023), Lorenza eliminates this overhead through a backpropagation-free perturbation (BPFP) scheme, significantly reducing computational and memory complexity. For example, for the OPT-13B model, the backpropagation consumes approximately ×6 more memory during fine-tuning than using the proposed BPFP and approximately ×4 more in calculation time. Furthermore, Lorenza employs a dynamic low-rank subspace selection mechanism, ensuring optimization updates remain efficient while maintaining the benefits of full-rank tuning. From a theoretical perspective, we establish a convergence guarantee for Lorenza, proving that it efficiently finds flat minima that promote better generalization while maintaining computational efficiency and significantly reducing memory overhead.

Finally, we empirically demonstrate that Lorenza outperforms state-of-the-art methods in pre-training and fine-tuning of LLMs. Notably, Lorenza exhibits robust adaptation across diverse datasets and challenging tasks, highlighting its potential as a scalable, efficient alternative for training and fine-tuning LLMs under resource constraints.

## 2 Related work

The generalization ability of neural networks has been shown to correlate with the flatness of the minima (Hochreiter & Schmidhuber, 1997; Keskar et al., 2017; Sun et al., 2023; Si & Yun, 2023; Yue et al., 2024). In regions around flat minima in the loss landscape, as illustrated in Figure 1, small parameter changes lead to minimal loss variation, reducing the model's sensitivity to noise and perturbations. This robustness has been shown to enhance the model's ability to generalize to unseen data, compared to standard optimization methods that may converge to sharp minima.

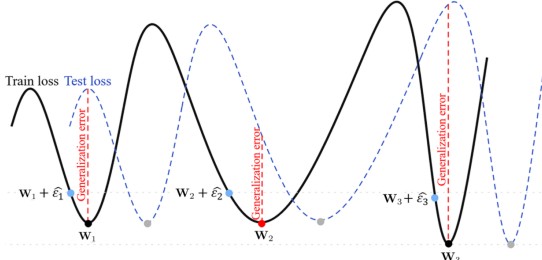

Figure 1: An illustration showing how the flatness of different minima can impact test loss. Specifically, $\mathbf{W}_1$ and $\mathbf{W}_3$, are located in sharp regions that have a high generalization error, while $\mathbf{W}_2$, found in a flatter region, exhibits a lower generalization error (Dong et al., 2024).

**Sharpness-Aware Minimization.** Sharpness-Aware Minimization (SAM) (Foret et al., 2021) aims to solve the min-max optimization problem: $\min_{\mathbf{w}} \max_{\|\boldsymbol{\epsilon}\|_2 \leq \rho} f_S(\mathbf{w} + \boldsymbol{\epsilon})$, where $f_S(\mathbf{w})$ denotes the empirical loss. Note that the objective value per $\mathbf{w}$ is equal to the highest value of the loss within a neighborhood of $\mathbf{w}$, defined as a ball of radius $\rho$ centered at $\mathbf{w}$. Therefore, this problem promotes flat minimizers, where small perturbations in the weights (even the "worst" $\boldsymbol{\epsilon}$) do not increase the empirical loss significantly.

To simplify the problem, SAM approximates the solution to the inner maximization using a first-order Taylor expansion around $\mathbf{w}$. This leads to the following approximation of the perturbation, $\boldsymbol{\epsilon} = \arg\max_{\|\boldsymbol{\epsilon}\|_2 \leq \rho} f_S(\mathbf{w} + \boldsymbol{\epsilon}) \approx \rho \frac{\nabla_{\mathbf{w}} f_S(\mathbf{w})}{\|\nabla_{\mathbf{w}} f_S(\mathbf{w})\|_2}$. Substituting this back into the outer minimization reformulates the objective as: $\min_{\mathbf{w}} f_S \left( \mathbf{w} + \rho \frac{\nabla_{\mathbf{w}} f_S(\mathbf{w})}{\|\nabla_{\mathbf{w}} f_S(\mathbf{w})\|_2} \right)$. In practice, given a mini-batch $B$, SAM extends the standard stochastic gradient descent (SGD) (Battash et al., 2024) update to the following two-step process: (1) for the current weights $\mathbf{w}_t$, compute the adversarial perturbation: $\boldsymbol{\epsilon}_t = \rho \frac{\nabla_{\mathbf{w}} f_B(\mathbf{w}_t)}{\|\nabla_{\mathbf{w}} f_B(\mathbf{w}_t)\|_2}$, (2) evaluate the gradient of the perturbed weights $\mathbf{w}_t + \boldsymbol{\epsilon}_t$ and use it to update $\mathbf{w}_t$, namely $\mathbf{w}_{t+1} = \mathbf{w}_t - \eta \mathbf{G}_t^{\text{SAM}}$, where $\mathbf{G}_t^{\text{SAM}} = \nabla_{\mathbf{w}} f_B(\mathbf{w}_t + \boldsymbol{\epsilon}_t)$, and $\eta$ is the learning rate. This procedure ensures that SAM balances the trade-off between minimizing the empirical loss and achieving a flat minimum, improving generalization performance.

**Variants of SAM.** AdaSAM (Sun et al., 2023) enhances SAM by integrating adaptive estimates of the first and second moments of the gradients to further improve optimization efficiency and generalization in deep neural networks, similar to the popular Adam optimizer (Kingma & Ba, 2017) and its weight decay regularization variant, AdamW (Loshchilov & Hutter, 2019). Specifically, the algorithm alternates between calculating a perturbation and updating parameters using the Adam optimization rule. Formally, AdaSAM modifies the SAM optimization process by incorporating the notion of AdaM (Tan et al., 2019), introducing a momentum term, $\mathbf{M}_t = \beta_1 \mathbf{M}_{t-1} + (1-\beta_1)\mathbf{G}_t^{\text{SAM}}$, which is weighted by a momentum factor $\beta_1$. Additionally, it tracks a second-moment estimate using a smoothing parameter $\beta_2$, namely $\mathbf{V}_t = \beta_2 \mathbf{V}_{t-1} + (1-\beta_2)\left[\mathbf{G}_t^{\text{SAM}}\right]^2$. This allows it to dynamically adjust using historical gradient information. It achieves a convergence rate of $\mathcal{O}(1/\sqrt{bT})$, where $b$ is the batch size, resulting in a linear speedup with increased batch sizes and making

it suitable for large-scale training scenarios. This adaptive variant requires the storage of both $\mathbf{M}_t$ and $\mathbf{V}_t$ at each time step, resulting in a memory cost of $2mn$. Additionally, the perturbation introduces an extra memory usage of $mn$, bringing the total memory access for AdaSAM optimization to $4mn$. It is important to note that this inefficiency leads to high memory requirements and increases computational time. Compared to gradient-based optimizers, SAM and its variants involve two gradients, which means two backpropagation procedures are performed during a single update step.

Surrogate Gap Guided Sharpness-Aware Minimization (GSAM) (Zhuang et al., 2022) jointly minimizes the perturbed loss $f_p(\mathbf{w}) := \max_{\|\boldsymbol{\epsilon}\| \leq \rho} f(\mathbf{w} + \boldsymbol{\epsilon})$ and the surrogate gap $h(\mathbf{w}) = f_p(\mathbf{w}) - f(\mathbf{w})$. It decomposes $\nabla f(\mathbf{w}) = \nabla_{\parallel} f(\mathbf{w}) + \nabla_{\perp} f(\mathbf{w})$, parallel and orthogonal to $\nabla f_p(\mathbf{w})$, and updates via $\mathbf{w}_{t+1} = \mathbf{w}_t - \eta \left( \nabla f_p(\mathbf{w}_t) - \alpha \nabla_{\perp} f(\mathbf{w}_t) \right)$, promoting flatter minima for better generalization.

**Memory efficient optimizers.** Reducing the memory demands of training large language models (LLMs) has driven extensive research in algorithmic development. One major approach reduces trainable parameters via low-rank adaptation (Hu et al., 2021), though such methods often fall short of fully parameterized models, especially during pre-training. Another direction focuses on optimizing training methods to reduce the memory footprint of optimizer states through low-rank gradient projection, with notable examples including AdaRankGrad, GaLore, Fira, Flora, Adam-mini, GaLore-mini, LDAdam, GoLore, LoQT, Apollo, and AdamSN, SubTrack++(Refael et al., 2025; Zhao et al., 2024; Chen et al., 2024b; Hao et al., 2024; Robert et al., 2025; Zhang et al., 2024a; Nguyen & Nguyen, 2025; Rajabi et al., 2025) integrating low-rank gradient projections in optimization. Efforts to reduce SAM's memory requirements have also been reported. They all appear to focus solely on the overhead caused by the perturbation (ascent step) computation. FSAM (Zhong et al., 2022) and SSAM (Zhao et al., 2023) leverage Fisher information to selectively perturb a parameter subset, achieving 50%-90% memory savings from the overhead at the cost of increased computation. Recent work on $\nu$-SAM (Anonymous, 2024) employs nuclear norm constraints during the ascent step for greater memory efficiency. Similarly, SAM-ON (Mueller et al., 2023) introduces perturbations only to the normalization layers. However, these approaches do not address the high memory demands of the baseline optimizer performing the descent step. Furthermore, they often trade memory savings related to the ascent step for increased computational complexity or struggle to generalize across diverse fine-tuning tasks. Our method bridges this gap by introducing a low-rank gradient optimization framework that is applied in both ascent and descent directions. We also estimate randomized ascent direction (gradient perturbation), leading to both memory efficiency and computational simplicity while enabling robust generalization in pre-training and fine-tuning LLMs.

**Low-rank gradient optimization.** The phenomenon of low-rank gradients naturally arises during the training of neural networks, a subject that has been extensively examined both theoretically and practically, e.g., (Zhao et al., 2022; Cosson et al., 2023; Yang et al., 2023). This characteristic low-rank structure gradient has been leveraged to reduce memory usage during training processes (Gooneratne et al., 2020; Huang et al., 2023; Modoranu et al., 2023), resulting in a reduced computational complexity compared to standard gradient descent methods. Recent work in (Refael et al., 2025) theoretically and empirically showed a natural phenomenon in which the rank of reversible layer gradients (Tian et al., 2021) monotonically diminishes to one during training and suggested leveraging it to adaptively reduce the rank of the gradients during Adam optimization steps.

**Zeroth-Order Optimization** Zeroth-order (ZO) optimization estimates gradients using finite differences and relies only on function value oracles. Despite this, its structure is similar to first-order (FO) gradient-based methods. It has gained significant attention due to its effectiveness across various modern machine learning challenges (Liu et al., 2020). Methods for ZO include approaches that leverage historical data to enhance gradient estimators (Meier et al., 2019; Cheng et al., 2021). These methods utilize gradient structural information (Singhal et al., 2023), exploit sparsity to reduce dimensional dependence (Cai et al., 2021; 2022; Chen et al., 2024a), and reuse intermediate features (Chen et al., 2024a) or random perturbations (Malladi et al., 2023). These strategies have demonstrated significant advancements in addressing large-scale machine learning challenges. In this study, we will further leverage the effectiveness of ZO to reduce unnecessary computational costs.

## 3 Method

In this section, we propose a computationally and memory-efficient adaptive-SAM optimization method that utilizes efficient zero-order gradient estimation to compute the ascent direction (perturbation) and analyze its convergence guarantees. Next, we leverage a computationally efficient subspace selection method, which offers lower complexity compared to an exact Singular Value Decomposition (SVD). This subspace selection is later used to determine the optimization subspace (gradient projection). Finally, we present Lorenza, along with its enhancement variant Lorenza+, which both apply memory and computationally efficient low-rank SAM optimization updates.

### 3.1 Single gradient SAM approach via zeroth-order ascent estimation

In this subsection, we introduce Algorithm 1, which we call AdaZo-SAM. This algorithm is designed to reduce computational effort, training time, and memory consumption associated with the adaptive SAM schema. Unlike traditional gradient-based optimizers, SAM and its variants require the calculation of two gradients, effectively performing backpropagation twice in a single update step. Inspired by the randomized gradient estimator (RGE) (Nesterov & Spokoiny, 2017; Duchi et al., 2015), which relies on the finite difference of function values along a randomly chosen direction vector, we propose estimating the perturbation or ascent direction instead of calculating the exact SAM perturbation. This approach eliminates the need for additional costly gradient calculations, allowing backpropagation to be applied only once during each update step rather than twice.

Given a scalar-valued function $f(\mathbf{W})$ where $\mathbf{W} \in \mathbb{R}^{m \times n}$, the RGE (referred to as $\hat{\nabla} f(\mathbf{W})$) is expressed using a central difference scheme, namely

$$\hat{\nabla} f(\mathbf{W}) = \frac{1}{q} \sum_{i=1}^{q} \left[ \frac{f(\mathbf{W} + \mu \mathbf{U}_i) - f(\mathbf{W} - \mu \mathbf{U}_i)}{2\mu} \mathbf{U}_i \right], \tag{1}$$

where $\mathbf{U}_i \in \mathbb{R}^{m \times n}$ is a random direction vector typically drawn from the standard Gaussian distribution $\mathcal{N}(\mathbf{0}, \mathbf{I})$, $q$ is the number of function queries, and $\mu > 0$ is a small perturbation stepsize (also known as smoothing parameter). The value of $q$ trades off the variance of the RGE gradient estimator against the overall computation cost. Notice that as $\mu \to 0$ and $q = 1$, the finite difference of the function values in approaches $f'(\mathbf{W}, \mathbf{U}_i) := \mathrm{Tr}\left(\nabla f(\mathbf{W})^\top \mathbf{U}_i\right) = \mathrm{vec}(\mathbf{U}_i)^\top \mathrm{vec}(\nabla f(\mathbf{W}))$, denoting the directional derivative of $f(\mathbf{W})$, along the random direction $\mathbf{U}_i$, yielding $\hat{\nabla} f(\mathbf{W}) \to f'(\mathbf{W}, \mathbf{U}_i) \mathbf{U}_i$, thus,

$$\lim_{\mu \to 0} \mathbb{E}\left[ \mathrm{vec}\left( \hat{\nabla} f(\mathbf{W}) \right) \right] = \mathbb{E}_{\mathbf{U}}\left[ \mathrm{vec}(\mathbf{U}_i)\, f'(\mathbf{W}, \mathbf{U}_i) \right]$$

$$= \mathbb{E}_{\mathbf{U}}\left[ \mathrm{vec}(\mathbf{U}_i)\, \mathrm{vec}(\mathbf{U}_i)^\top \mathrm{vec}(\nabla f(\mathbf{W}_t)) \right] = \mathbb{E}\left[ \mathrm{vec}(\nabla f(\mathbf{W}_t)) \right]. \tag{2}$$

It might be assumed that estimating the ascent direction would break the convergence guarantees of SAM and AdamSAM. However, we show that even using just one randomized matrix, i.e., $q = 1$, is sufficient to maintain the same convergence properties. To analyze the convergence of the AdaZo-SAM algorithm, for the simplicity of writing, we let $\mathbb{E}_\xi[\nabla f(\mathbf{W})] = \mathbb{E}_\xi[\nabla_\mathbf{W} f(\mathbf{W}; \xi)]$, where $\xi \sim \mathbb{P}_\mathcal{D}$ is a stochastic input batch, and $\mathbb{P}_\mathcal{D}$ is the sampling distribution over dataset/domain $\mathcal{D}$.

**Theorem 3.1** (AdaZo-SAM convergence rate). *Consider a $\beta$-smooth, non-convex function $f$ parametrized by a matrix $\mathbf{W} \in \mathbb{R}^{m \times n}$, where $m \leq n$ without loss of generality. Suppose that $f$ satisfies $\sup_{\mathbf{W}} \mathbb{E}_\xi \| f(\mathbf{W}; \xi) \| \leq C$ for some large constant $C \in \mathbb{R}_+$. Then, Algorithm 1, when initialized at $\mathbf{W}_0$ and run with step size $\eta = \frac{1}{\beta \sqrt{T}}$, achieves the convergence rate*

$$\frac{1}{T} \sum_{t=0}^{T-1} \mathbb{E} \left\| \hat{\nabla} f(\mathbf{W}_t) \right\|_F^2 \leq \mathcal{O}\left( \frac{C\beta}{\sqrt{T}} \right) + \beta^2 \rho^2,$$

---

**Algorithm 1** Efficient single gradient step Adaptive SAM with zeroth-order ascent estimation (AdaZo-SAM)

---

Initial parameters $\mathbf{W}_0, \mathbf{M}_{-1} = \mathbf{V}_{-1} = 0$, base learning rate $\gamma$, neighborhood size $\mu$, perturbation size $\rho$, and momentum parameters $\beta_1, \beta_2$, small number $q \in \mathbb{N}$ (default $q = 1$).
**Output:** Optimized parameter $\mathbf{W}_{T+1}$.
**for** iteration $t \in \{0, 1, 2, \ldots, T-1\}$ **do**
$\quad$ Sample mini-batch $B = \{\xi_1, \xi_2, \ldots, \xi_{|B|}\}$
$\quad$ (1) Random $\mathbf{U}_i \sim \mathcal{N}(\mathbf{0}, \mathbf{I}), i \in [q]$ $\qquad\qquad\qquad$ { Compute ascent direction (perturbation)}
$\quad$ (2) Compute $\mathbf{G}_t^{\text{Pert}}$ using (1): $\frac{1}{q} \sum_{i=1}^q \left[ \frac{-f(\mathbf{W}+\mu\mathbf{U}_i;\xi_t)+f(\mathbf{W}-\mu\mathbf{U}_i;\xi_t)}{2\mu} \mathbf{U}_i \right]$
$\quad$ Compute SAM gradient: $\mathbf{G}_t^{\text{SAM}} = \nabla_{\mathbf{W}} f(\mathbf{W}_t + \rho \frac{\mathbf{G}_t^{\text{Pert}}}{\|\mathbf{G}_t^{\text{Pert}}\|_F}; \xi_t)$
$\quad$ $\mathbf{M}_t = \beta_1 \mathbf{M}_{t-1} + (1 - \beta_1) \mathbf{G}_t^{\text{SAM}}$ $\qquad\qquad\qquad\qquad$ {Update momentum}
$\quad$ $\mathbf{V}_t = \beta_2 \mathbf{V}_{t-1} + (1 - \beta_2) (\mathbf{G}_t^{\text{SAM}})^2$ $\qquad\qquad\qquad$ {Update momentum and variance}
$\quad$ $\hat{\mathbf{M}}_t = \mathbf{M}_t / (1 - \beta_1^t)$
$\quad$ $\hat{\mathbf{V}}_t = \mathbf{V}_t / (1 - \beta_2^t)$
$\quad$ Update parameters:
$\quad$ $\mathbf{W}_{t+1} = \mathbf{W}_t - \gamma \hat{\mathbf{M}}_t / \left( \sqrt{\hat{\mathbf{V}}_t} + \epsilon \right)$
**end for**

---

where $\hat{\nabla} f(\mathbf{W}_t)$ denotes the randomized gradient estimator (RGE) defined in Equation (1), with $q = 1$, $\mu \to 0$, and $\xi \sim \mathbb{P}_\mathcal{D}$ sampled from the data distribution $\mathbb{P}_\mathcal{D}$ over domain $\mathcal{D}$.

The proof of Theorem 3.1 can be found in Appendix B.1. Consistent with all convergance proofs of other SAM variants (e.g., Si & Yun, 2023; Sun et al., 2023), Theorem 3.1 demonstrates that SAM converges to stationary points for non-convex smooth functions, provided the perturbation size $\rho$ is sufficiently small or decaying. Practically, for example, GSAM proposes scheduling $\rho_t$ as $\rho_t = \rho_{\min} + \frac{(\rho_{\max}-\rho_{\min})(lr-lr_{\min})}{lr_{\max}-lr_{\min}}$, enabling the algorithm to initially explore flatter regions of the loss landscape and then gradually reduce the perturbation size in proportion to the learning rate.

Recently, (Zhao et al., 2024; Refael et al., 2025) analyzed the gradient structure of a broad family of nonlinear networks known as *reversible networks*[1] (Tian et al., 2021). This family encompasses various types of layers, including linear layers (MLP and convolutional) and (leaky) ReLU nonlinearities. It was proved that the low-rank structure of their gradients naturally diminishes during training and fine-tuning, a phenomenon observed empirically across various layer types. This insight inspired the development of optimization methods that leverage low-rank update steps to reduce memory usage while enhancing accuracy, an approach we build upon and detail in the following subsection..

### 3.2 Low-rank gradient optimization via efficient zeroth-order adaptive SAM

Consider a neural network denoted as $\Phi(\cdot; \boldsymbol{\theta})$, which consists of $L$ layers and is parameterized by $\boldsymbol{\theta} \triangleq \left[ \mathbf{W}_1^{d_1 \times d_0}, \ldots, \mathbf{W}_{L-1}^{d_{L-1} \times d_{L-2}}, \mathbf{W}_L^{d_L \times d_0^{L-1}} \right]$. Here, $\mathbf{W}_i$ represents the weights tensor parameters associated with the $i$-th layer, for $i \in [L]$. By $f(\boldsymbol{\theta}; \xi)$ we denote the loss $\mathcal{L}$ for a sample $\xi$ and prediction $\Phi(\xi; \boldsymbol{\theta})$. With a slight abuse of notation, we write $f(\mathbf{W}; \xi)$ if the context refers to the weights of a certain layer. The proposed algorithm, Lorenza, is stated in Algorithm 2. The mathematical formulation of the weights update rule proposed in this paper is detailed in Appendix A.1. It comprises four main blocks, all contained within an outer loop that terminates when convergence is reached. The role of each block is as follows.

**Block 1**: We select the subspace along the directions of the $r$ largest eigenvectors, but since computing full SVD for large matrices is computationally intensive and memory-demanding, we leverage the Randomized-SVD by (Halko et al., 2010), and presented in Appendix 3, which is an efficient technique for producing a "good" proxy for the optimal low-rank approximation. It solves the optimization prob-

---

[1]These networks are formally defined in Appendix C.1.

lem $\arg\min_{\mathbf{Q}\in\mathbb{R}^{n\times r}}\left\|\mathbf{G}-\mathbf{QQ}^\top\mathbf{G}\right\|_F$, and approximates the matrix $\mathbf{G}$ as $\mathbf{G}_{\mathsf{app},r}\approx\mathbf{QQ}^\top\mathbf{G}$, that requires $O(mnr+mr^2)$ operations, instead of $O(\min(mn^2,m^2n))$ applied by SVD.

**Block 2**: We calculate the low-rank-perturbation representing the adversarial (ascent) direction within a low-rank subspace. This is achieved by randomizing a linear combination of directions within the subspace, projecting it back onto the original space, and then using it to empirically estimate the projected gradient on the selected subspace (i.e., the gradient components that reside in the selected low-rank subspace). Note that throughout the paper, we use $q=1$.

**Block 3**: The SAM direction is calculated using the low-dimensional and memory-efficient projected gradients on the selected subspace. It then updates Adam's estimates for the first and second moments.

**Block 4**: The low-rank Adam step is projected back onto the full dimensions, and the model parameters are updated until the convergence criteria are met. Such criteria could be, for example, the number of epochs or gradient norm reaching below a predefined threshold, namely $\|\mathbf{G}_t\|_F^2\leq\varepsilon$.)

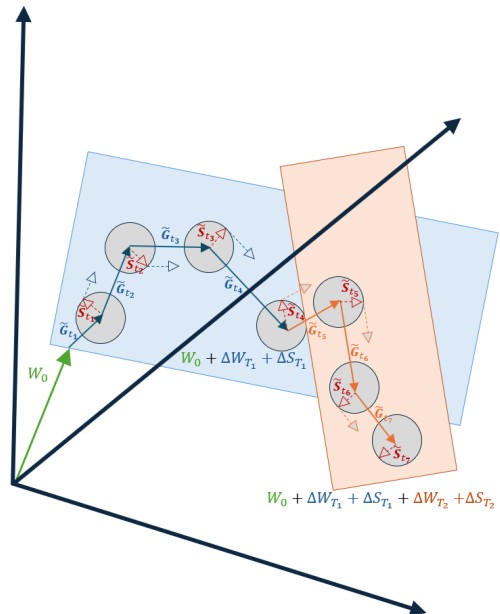

Figure 2: The illustration depicts the training process of Lorenza (2). The process begins by selecting a low-rank subspace using the efficient Randomized-SVD, visualized here as a 2D plane (blue and orange). Next, a low-rank AdaZo-SAM optimization step (1) is performed. Specifically, the estimated low-rank ascent direction $\tilde{\mathbf{S}}_t$ is computed using the RGE method, on the 2D-subspace. This low-rank ascent direction is being used to calculate the adversarial gradient $\mathbf{G}_t$, at the perturbated weights, $\mathbf{W}_t+\rho\frac{\tilde{\mathbf{S}}_t}{\|\tilde{\mathbf{S}}_t\|_2}$, then projected onto the 2D-subspace, namely as $\hat{\mathbf{G}}_t^{2\times m}=\mathbf{Q}_t^{2\times n}\tilde{\mathbf{G}}_t^{n\times m}$. Following this, a low-rank Adam optimization step is applied. After a predetermined number of Lorenza steps, the optimization subspace $\mathbf{Q}_t$ is updated, and the process is repeated.

**Theorem 3.2** (Convergence of Lorenza). *Suppose the following conditions hold:*

1. *The objective $f\equiv\mathcal{L}(\Phi(\cdot))$ is $\beta$-smooth, non-convex, and bounded by $M\in\mathbb{R}_+$.*

2. *Time intervals $T_\ell$ are determined by the criterion $\|\hat{\mathbf{G}}_{\mathsf{T}_\ell}\|\leq\varsigma_\ell$ for $\ell\in\mathbb{N}$ (subspace/low-rank projection update criteria).*

3. *Algorithm 2 utilizes vanilla SGD weight updates with exact SVD projection and a decay perturbation $\rho$.*

*Then, for any $\varepsilon > 0$, there exists a constant $\mathsf{C} \in \mathbb{R}_+$ and $N \in \mathbb{N}$ such that for all $\mathsf{T}_N > \frac{\mathsf{C}}{\varepsilon^2}$:*

$$\frac{1}{\mathsf{T}_N} \sum_{i=0}^{N-1} \sum_{t=\mathsf{T}_i}^{\mathsf{T}_{i+1}-1} \left\| \mathbf{G}_t^{j,SAM} \right\|_F^2 \leq \varepsilon. \tag{3}$$

*Specifically, Lorenza achieves an $\varepsilon$-critical point (i.e., $\|\mathbf{G}_t^j\|_F^2 \leq \varepsilon$) for some $t \in \mathbb{N}$ and all layers $j \in [L]$.*

The proof of Theorem 3.1 can be found in Appendix B.1. Several important points should be noted. First, to reduce memory usage, Algorithm 2 updates weights on a per-layer basis during backpropagation, following recent approaches (e.g., (Lv et al., 2024)). This differs from standard optimizers, which typically store full gradients in memory before updating all weights, leading to inefficiencies. Second, the Adam update step in Algorithm 2 can be replaced with any quantized Adam variant (e.g., (Li et al., 2017; Chen et al., 2021; Seok & Kim, 2021)), enabling fine-tuned quantized models or quantized adapters. Finally, 4-bit projected

---

**Algorithm 2** Lorenza / Lorenza+ optimization

---

**Input:** A weight matrix $\mathbf{W} \in \mathbb{R}^{m \times n}$ with $m \leq n$. Step size $\eta$, scale factor $\alpha$, decay rates $\{\beta_1, \beta_2\}$, weight decay $\lambda$, rank $r$, subspace update frequency $T$, small number $q \in \mathbb{N}$ (default $q = 1$), a small interval length $\mu$.

**Initialize:** $t \leftarrow 0$ and optionally, $\rho_t$ schedule: $\rho_t = \rho_{\min} + \frac{(\rho_{\max} - \rho_{\min})(lr - lr_{\min})}{lr_{\max} - lr_{\min}}$

**repeat**

  # Block 1: Calculate low rank gradient projection.

  Sample mini-batch $B = \{\xi_1, \xi_2, \ldots, \xi_{|B|}\}$

  Compute $\mathbf{G}_t \leftarrow \sum_{i=1}^{|B|} \frac{\partial}{\partial \mathbf{W}} \mathcal{L}(\Phi(x_i, \boldsymbol{\theta}), y_i)$

  **if** $t \bmod K = 0$ **then**

    $\mathbf{Q}_t \leftarrow \text{Truncated\_Randomized\_SVD}(\mathbf{G}_t)$

  **end if**                               {Alternatively criteria $\|\hat{\mathbf{G}}_t\| \leq \varsigma$}

  # Block 2: Low-rank rank ascent perturbation

  Randomize vector $\mathbf{u}_j^{r \times 1} \sim \mathcal{N}(\mathbf{0}, 1)$, $j \in [q]$          {Compute low-rank random directions}

  Set $\mathbf{P}_j = \mathbf{Q}_t \text{diag}(\mathbf{u}_j) \mathbf{R}_t$, $j \in [q]$

  $\mathbf{G}_t^{\text{Pert}} = -\frac{1}{q} \sum_{\xi_i \in B, j \in [q]} \left[ \frac{f(\mathbf{W}_t + \mu \mathbf{P}_j; \xi_i) - f(\mathbf{W}_t - \mu \mathbf{P}_j; \xi_i)}{2\mu} \mathbf{P}_j \right]$    {Compute ascent direction (perturbation)}

  # Block 3: Low-rank adaptive SAM

  $\mathbf{G}_t^{\text{SAM}} = \frac{1}{|B|} \sum_{\xi_i \in B} \frac{\partial}{\partial \mathbf{W}} f\left( \mathbf{W}_t + \rho \frac{\mathbf{G}_t^{\text{Pert}}}{\|\mathbf{G}_t^{\text{Pert}}\|_F}; \xi_i \right)$           {Compute decent direction}

  $\hat{\mathbf{G}}_t \longleftarrow \mathbf{Q}_t^\top \mathbf{G}_t^{\text{SAM}}$

  $\mathbf{M}_t \longleftarrow \beta_1 \mathbf{M}_t + (1 - \beta_1) \hat{\mathbf{G}}_t$

  $\mathbf{V}_t \longleftarrow \beta_2 \mathbf{V}_t + (1 - \beta_2) \hat{\mathbf{G}}_t^2$

  $\hat{\mathbf{M}}_t \longleftarrow \mathbf{M}_t / (1 - \beta_1^t)$

  $\hat{\mathbf{V}}_t \longleftarrow \mathbf{V}_t / (1 - \beta_2^t)$

  # Block 4: Update weight in original space.

  $\mathbf{W}_t \longleftarrow \mathbf{W}_t - \alpha \mathbf{Q}_t \hat{\mathbf{M}}_t / \left( \sqrt{\hat{\mathbf{V}}_t} + \epsilon \right)$           {Lorenza update step}

  $\mathbf{G}_t^{\text{SAM}\perp} = \mathbf{G}_t^{\text{SAM}} - \mathbf{Q}_t \mathbf{Q}_t^\top \mathbf{G}_t^{\text{SAM}}$

  $\mathbf{W}_t \longleftarrow \mathbf{W}_t - \alpha \left( \frac{\mathbf{Q}_t \hat{\mathbf{M}}_t}{\sqrt{\hat{\mathbf{V}}_t} + \epsilon} + \mathbf{G}_t^{\text{SAM}\perp} \right)$           {Lorenza+ update step}

  $t \leftarrow t + 1$

**until** convergence criteria met                         {e.g. epoch number, gradient norm}

**return** $\mathbf{W}_T$                                     {A flat local minima}

---

gradient updates, as introduced in Q-GaLore (Zhang et al., 2024b), can be easily incorporated.

*Remark* 3.3. Our convergence statements (Theorem 3.1 and Theorem 3.2) assume a uniform bound on the loss values, e.g., $\sup_W \mathbb{E}_\xi \|f(W; \xi)\| \leq C$ (and similarly that $f$ is bounded by $M$, where $f \equiv \mathcal{L}(\Phi(\cdot))$ is the

composition of loss $\mathcal{L}$ and the network $\Phi$). This assumption plays a technical role in the proofs: since Lorenza/AdaZo-SAM uses zeroth-order finite-difference function queries, boundedness of $f(W; \xi)$ controls these evaluations and ensures the resulting estimator is well-behaved in expectation, which is needed for the stated rate. The assumption is reasonable for the regime of interest: training is performed on a finite dataset and the iterates $\{W_t\}$ produced by the algorithm are effectively confined to a bounded (hence compact) region; by continuity, the empirical loss is bounded on this region by some finite constant $M$, consistent with our assumption. In classification and language modeling with cross-entropy, the loss is nonnegative and remains finite along the sequence $\{W_t\}$ (in particular, the logits do not diverge). Moreover, standard practices such as weight decay and normalization layers (e.g., LayerNorm) further restrict the effective search region, making unbounded loss growth unlikely. Finally, bounded-loss-type conditions also appear in analyses of memory-efficient low-rank optimization frameworks, and we adopt them here to obtain a stronger guarantee in our zeroth-order/low-rank SAM setting. We emphasize that, although this assumption is stronger than the smoothness/bounded-variance conditions commonly adopted in related work (e.g., Si & Yun (2023); Sun et al. (2023)), Theorems 3.1 and 3.2 also provide a correspondingly stronger guarantee. However, this assumption is standard in convergence analyses of low-rank optimization methods; see, for example, the convergence proof of GaLore in Zhao et al. (2024).

*Remark* 3.4 (Memory usage). Our goal is to reduce end-to-end training memory, i.e., both (i) optimizer-state memory and (ii) SAM-specific overhead. First, standard Adam requires storing gradients in addition to first- and second-moment buffers, which can be up to $\sim 3\times$ the parameter memory. In contrast, Lorenza keeps the same model weights in memory but replaces full-rank optimizer states with low-rank states, so the optimizer-state footprint scales with the rank $r$ rather than the matrix dimension $mn$. This is summarized in Table 1: for $W \in \mathbb{R}^{n \times m}$, Lorenza's optimizer states scale as $nr + 2mr$, comparable to GaLore and substantially smaller than SAM/AdaSAM, which maintain full-rank states and (for SAM variants) incur additional overhead due to two backward passes. Second, to reduce optimizer-state memory, Lorenza is designed to reduce *peak* GPU memory usage, which is typically dominated by the largest set of tensors that must be simultaneously resident during an update (activations/gradients, plus any additional SAM buffers). Standard SAM/AdaSAM incur a larger peak footprint because they require *two* backpropagations per step, effectively repeating activation/gradient materialization and adding SAM-specific overhead. By contrast, our BPFP construction enables *one* backpropagation per step (Table 1), which directly reduces this peak requirement; empirically, for OPT-13B, the backprop-based perturbation computation can consume $\sim 6\times$ more memory during fine-tuning than BPFP. Finally, Algorithm 2 further reduces peak usage by updating weights *per-layer during backpropagation*, avoiding the need to store full gradients for all layers before applying updates, as proposed by recent works, see, e.g., Lv et al. (2024).

## 3.3 Lorenza+: Enhancement of Lorenza

We introduce **Lorenza+**, an extension of the Lorenza algorithm (Algorithm 2) that better exploits previously computed gradient information, thereby improving performance across a wide range of architectures.

In the standard Lorenza update, the optimizer discards the component of the SAM gradient $\mathbf{G}_t^{\mathrm{SAM}}$ that lies outside the low-rank subspace spanned by $\mathbf{Q}_t$, i.e., $\mathbf{G}_t^{\mathrm{SAM}\perp} = \mathbf{G}_t^{\mathrm{SAM}} - \mathbf{Q}_t \mathbf{Q}_t^\top \mathbf{G}_t^{\mathrm{SAM}}$. This orthogonal term captures informative directions that are not represented in the low-rank subspace. Importantly, it does not interfere with the moment estimates, meaning it can be incorporated without compromising stability. Yet, existing optimizers such as GaLore, Flora, Adam-mini, GaLore-mini, LDAdam, AdamSN, and SubTrack++ ignore this orthogonal component in their update rules.

Lorenza+ leverages this neglected term. Specifically, it augments the Lorenza update with an additional SGD-like correction that uses $\mathbf{G}_t^{\mathrm{SAM}\perp}$, while retaining the original low-rank update through $\mathbf{M}_t$ and $\mathbf{V}_t$. The resulting update rule is, $\mathbf{W}_t \leftarrow \mathbf{W}_t - \alpha \left( \frac{\mathbf{Q}_t \hat{\mathbf{M}}_t}{\sqrt{\hat{\mathbf{V}}_t} + \epsilon} + \mathbf{G}_t^{\mathrm{SAM}\perp} \right)$. Here, the first term coincides with the standard Lorenza step, while the second introduces the complementary gradient component. Since $\mathbf{G}_t^{\mathrm{SAM}}$ is already computed and stored in each iteration, no extra memory is required. Furthermore, because $\mathbf{Q}_t$ is of a low rank (typically rank 4, 8, or 16), the additional computational overhead is negligible.

## 4   Experiments

**Fine-tuning on the GLUE Benchmark.**   We evaluate our approach on the GLUE benchmark (Wang et al., 2019) by fine-tuning the pre-trained RoBERTa-base model (Liu et al., 2019). The results, compared against full fine-tuning, LoRA, and GaLore methods, are summarized in Table 2. For evaluation metrics, we report overall accuracy (matched and mismatched) for MNLI, Matthew's correlation for CoLA, Pearson correlation for STS-B, F1-score for MRPC, and accuracy for the remaining tasks. Our method demonstrates improved fine-tuning accuracy while maintaining comparable training memory on average. We employ $\rho$ scheduling (as proposed in the GSAM method (Zhuang et al., 2022)), with $\rho_{max} = 0.01$, $\rho_{min} = 1e - 6$, and a cosine annealing learning rate scheduler.

Table 2: Evaluating Lorenza and Lorenza+, compared to state-of-the-art memory-efficient fine-tuning methods on the GLUE benchmark using pre-trained RoBERTa-Base. We used NVIDIA A100 for the experiments.

| Model | Memory | CoLA | STS-B | MRPC | RTE | SST2 | MNLI | QNLI | QQP |
|---|---|---|---|---|---|---|---|---|---|
| Full Fine-Tuning | 747M | 62.24 | 90.92 | 91.30 | 79.42 | 94.57 | 87.18 | 92.33 | 92.28 |
| $\nu$ SAM | > 747M | 57.38 | 87.43 | 90.46 | 59.93 | 92.68 | 84.16 | 90.11 | 91.29 |
| SAM-ON | > 747M | 57.84 | 87.41 | 89.95 | 60.58 | 92.27 | 83.98 | 90.90 | 91.29 |
| LoRA (rank=4) | 257M | 61.38 | 90.57 | 91.07 | 78.70 | 92.89 | 86.82 | 92.18 | 91.29 |
| GaLore (rank=4) | 253M | 60.35 | 90.73 | 92.25 | 79.42 | 94.0 | 87.0 | 92.24 | 91.06 |
| **Lorenza** (rank=4) | 253M | 61.51 | 91.01 | 92.57 | 81.26 | 94.63 | **87.2** | 92.54 | 91.82 |
| **Lorenza+** (rank=4) | 253M | **61.97** | **91.14** | **92.76** | **81.35** | **94.68** | 87.16 | **92.78** | **91.88** |
| LoRA (rank=8) | 264M | 61.83 | 90.80 | 91.90 | 79.06 | 93.46 | 86.94 | 92.25 | 91.22 |
| GaLore (rank=8) | 257M | 60.06 | 90.82 | 92.0 | 79.78 | 94.38 | 87.17 | 92.2 | 91.11 |
| LDAdam (rank=8) | 257M | 59.91 | 90.11 | 88.40 | 78.96 | **95.00** | **87.81** | 92.87 | 91.28 |
| SubTrack++ (rank=8) | 258M | 58.29 | 90.90 | 91.84 | 76.53 | 90.83 | – | – | – |
| BAR (Li et al., 2024) | 253M | 62.14 | 91.07 | 92.61 | 81.0 | 94.18 | 87.46 | 92.36 | **91.6** |
| **Lorenza** (rank=8) | 257M | 62.1± 0.019 | 90.93± 0.007 | 92.8± 0.022 | 81.17± 0.017 | 94.84± 0.023 | 87.14± 0.017 | 92.72± 0.029 | 91.26± 0.011 |
| **Lorenza+** (rank=8) | 257M | **62.31**± 0.027 | **91.18**± 0.023 | **92.93**± 0.026 | **81.26**± 0.019 | 94.84± 0.013 | 87.23± 0.021 | **92.85**± 0.036 | 91.38± 0.014 |

**Pre-training LLAMA on C4 Dataset.**   To evaluate the performance of Lorenza, we follow (Zhao et al., 2024) by comparing it with the state-of-the-art method in terms of perplexity and memory efficiency. For this evaluation, we trained large LLaMA-based models on the C4 dataset, a curated and extensive version of the Common Crawl web corpus (Raffel et al., 2020). This dataset is widely used for pre-training language models and developing word representations. To better reflect real-world pre-training scenarios, we conducted training on a non-repeating, large-scale dataset and scaled model sizes up to 1 billion parameters. The results of these experiments are shown in Table 3.

Table 3: Comparison of low-rank state-of-the-art algorithms for **pre-training** LLaMA models of varying sizes on the C4 dataset. The results are reported in terms of validation perplexity. Experiments were conducted using an NVIDIA H200 GPU.

| Method | 60M | 130M | 350M | 1B |
|---|---|---|---|---|
| Full-Rank | 34.06 (0.36G) | 25.08 (0.76G) | 18.80 (2.06G) | 15.56(7.80G) |
| GaLore | 34.88 (0.24G) | 25.36 (0.52G) | 18.95 (1.22G) | 15.64(4.38G) |
| Low-Rank | 78.18 (0.26G) | 45.51 (0.54G) | 37.41 (**1.08**G) | 142.53(3.57G) |
| LoRA | 34.99 (0.36G) | 33.92 (0.80G) | 25.58 (1.76G) | 19.21(6.17G) |
| ReLoRA | 37.04 (0.36G) | 29.37 (0.80G) | 29.08 (1.76G) | 18.33(6.17G) |
| AdamSN | 29.75 (0.36G) | 22.90 (0.80G) | 17.49 (1.76G) | 14.96(6.17G) |
| Apollo | 31.55 (0.26G) | 22.94 (0.57G) | 16.85 (1.29G) | 14.20 (4.43G) |
| Apollo mini | 31.93 (0.23G) | 23.53 (0.43G) | 17.18 (0.93G) | 14.17 (2.98G) |
| LDAdam | – | 22.65 (0.54G) | 17.30 (1.23G) | - |
| **Lorenza** | 34.29 (0.24G) | 24.92 (0.53G) | 18.87 (1.24G) | 14.93 (4.38G) |
| **Lorenza+** | **29.69** (0.24G) | **22.38** (0.53G) | **16.94** (1.24G) | **13.98** (4.38G) |
| Training Tokens | 1.1B | 2.2B | 6.4B | 13.1B |
| $r/d_{\text{model}}$ | 128/256 | 256/768 | 256/1024 | 512/2048 |

**Few/Zero-shot reasoning and long-context generalization.**   To evaluate the performance of our method on a complex reasoning task, we utilize the GSM8K dataset (Cobbe et al., 2021) to test systematic

generalization. For these experiments, we used a batch size of 32 and 10 epochs for fine-tuning. We present the performance result in Table 4 training Phi-2 (**2.7B**) model (Javaheripi et al., 2023), and training Lamma (1B) model (Touvron et al., 2023). The results demonstrate that the proposed method significantly improves generalization to out-of-distribution data. The experiments were conducted on an NVIDIA H200.

Table 4: Evaluation on GSM8K: Zero-shot (Phi-2) and 8-shot (LLaMA) settings, both with Rank= 64.

| **Zero-shot (Phi-2, 2.7B)** | Model | Accuracy | | **8-shot (LLaMA, 1B)** | Model | Accuracy |
|---|---|---|---|---|---|---|
| | Base | 15.16% | | | Base | 17.93% |
| | Galore | 52.24% | | | Galore | 74.9% |
| | LoRA | 42.8% | | | LoRA | 68.3% |
| | LDAdam | 52.47% | | | LDAdam | 75.29% |
| | **Lorenza** | 53.37% | | | **Lorenza** | 76.4% |
| | **Lorenza+** | **55.26**% | | | **Lorenza+** | **77.1**% |

In addition, Table 5 presents the results of pretraining an LLaMA-350M model, where we used a sequence length of 1024 and a total batch size of 512 for 60,000 steps. For the baseline AdamW optimizer, we conducted a learning rate sweep and selected the best checkpoint. Our Lorenza method was evaluated under the same conditions for consistency. We follow the zero-shot evaluation protocol described in Table 4 of Zhu et al. (2025), using a comprehensive suite of commonsense and mathematical reasoning benchmarks. Further details are specified in Appendix D.

Table 5: Zero-shot evaluation of LLaMA-350M models across reasoning tasks.

| Method | Memory | PPL | BoolQ | RTE | HS | WG | OBQA | ARC-E | ARC-C | PIQA | SciQ | MathQA | Avg. |
|---|---|---|---|---|---|---|---|---|---|---|---|---|---|
| AdamW | 1.37G | 16.30 | 0.4917 | 0.4693 | 0.3688 | 0.5233 | 0.332 | 0.3729 | 0.2449 | 0.6534 | 0.609 | 0.2064 | 0.4272 |
| APOLLO | 0.34G | 15.64 | 0.5373 | 0.4698 | 0.3850 | 0.4925 | 0.322 | 0.3788 | 0.2483 | 0.6681 | 0.624 | 0.2127 | 0.4406 |
| APOLLO-Mini | 0.15G | 16.12 | 0.5376 | 0.4562 | 0.3707 | 0.5217 | 0.324 | 0.3758 | 0.2312 | 0.6638 | 0.619 | 0.2224 | 0.4374 |
| **Lorenza** | 0.31G | 15.53 | 0.5450 | 0.4702 | 0.3900 | 0.5250 | **0.328** | 0.3840 | 0.2510 | 0.6695 | 0.627 | 0.2240 | 0.4413 |
| **Lorenza+** | 0.31G | **15.27** | **0.5481** | **0.4711** | **0.3924** | **0.5318** | 0.327 | **0.3846** | **0.2521** | **0.6703** | **0.628** | **0.2265** | **0.4432** |

Table 6: **The LLaMA 7B model** was pre-trained on the C4 dataset for 120K steps with **8-bit quantization** range in optimization. Lorenza and Galore were utilized for low-rank training, specifically with a rank of 256. Experiments were conducted using an NVIDIA H200.

| Steps/Tokens | GaLore | Adam | Lorenza | Lorenza+ |
|---|---|---|---|---|
| 40K / 5.2B | 17.94 | 18.09 | **17.81** | 18.03 |
| 80K /10.5B | 15.39 | 15.47 | 15.28 | **15.11** |
| 120K /15.7B | 14.95 | 14.83 | 14.82 | **14.78** |

The following hyper-parameters were used in the experiments: $\alpha = 4$, WD $= 0.5 \times 10^{-4}$, $T = 180$, $\mu = 10^{-9}$, $q = 1$, $\varepsilon = 10^{-12}$, lr $= 0.5 \times 10^{-3}$, $\rho_{\max}/\rho_{\min} = 10^{-1}/10^{-6}$.

## 4.1 Ablation Studies

### 4.1.1 Block 1: Effect of Using Full SVD in Projection

In Block 1 of the Lorenza optimizer, the projection matrix is computed using a randomized SVD, which serves as a practical approximation to reduce computational costs associated with a full SVD. To assess the trade-off, we conduct an ablation study comparing the original randomized SVD variant with one that uses the exact full SVD computation.

The results show that replacing the randomized SVD with a full SVD yields only marginal performance improvements, while incurring a substantial training time overhead of approximately 30%. The memory usage difference is negligible and thus not reported.

Table 7: Ablation Study: Randomized SVD vs. Full SVD in Block 1

| Model | CoLA | STS-B | MRPC | RTE | SST2 | MNLI | QNLI | QQP |
|---|---|---|---|---|---|---|---|---|
| Lorenza (Rank=4) | 62.1 | 90.93 | 92.8 | 81.17 | 94.84 | 87.14 | 92.72 | 91.26 |
| Using Full-SVD | 62.6 | 90.98 | 93.04 | 81.24 | 94.91 | 87.27 | 92.86 | 91.32 |
| Added Time Ratio | ×1.37 | ×1.28 | ×1.34 | ×1.32 | ×1.34 | ×1.31 | ×1.36 | ×1.29 |

#### 4.1.2 Block 2: Effect of Backpropagation in Low-Rank Gradient Estimation

Block 2 approximates the low-rank gradient using a zero-order (backpropagation-free) method, thereby avoiding the computational burden of full backward passes. This ablation study examines the effect of replacing this approximation with the exact gradient obtained through backpropagation.

Table 8: Ablation Study: Zero-Order vs. Backpropagation-Based Gradient in Block 2

| Model | CoLA | STS-B | MRPC | RTE | SST2 | MNLI | QNLI | QQP |
|---|---|---|---|---|---|---|---|---|
| Lorenza (Rank=4) | 62.1 | 90.93 | 92.8 | 81.17 | 94.84 | 87.14 | 92.72 | 91.26 |
| Using Backpropagation | 62.4 | 91.07 | 93.16 | 81.03 | 94.92 | 87.22 | 92.94 | 91.23 |
| Added Time Ratio | ×1.23 | ×1.27 | ×1.19 | ×1.21 | ×1.24 | ×1.26 | ×1.23 | ×1.21 |

The results indicate that while using true backpropagation slightly improves performance, the gains are minimal and come at the cost of a 20–25% increase in training time. As with Block 1, the memory usage differences are considered negligible.

## 5 Discussion

In this work, we introduce Lorenza, a novel memory-efficient optimization framework that enhances the generalization of low-resource training for large language models (LLMs). By combining zeroth-order sharpness-aware minimization with low-rank gradient updates, Lorenza reduces computational overhead while retaining the benefits of full-rank fine-tuning. Lorenza and Lorenza+, its enhanced version, achieve state-of-the-art performance across multiple benchmarks, surpassing existing low-rank adaptation methods such as LoRA and GaLore.

Our theoretical analysis further provides convergence guarantees, underscoring the robustness of our method. Lorenza effectively bridges the gap between memory efficiency and generalization, making it a strong alternative for resource-constrained scenarios. By addressing key limitations in PEFT, Lorenza enables efficient and generalizable fine-tuning, advancing LLM training and deployment.

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

# A Appendix

## A.1 Additional algorithm details

**Definition A.1.** [Low-Rank Gradient by Efficient Zeroth-Order Adaptive SAM (Lorenza)] Lorenza defines the following gradient update rules.

$$
\textbf{Lorenza}
\begin{cases}
\mathbf{G}_t^{\text{Pert}} = \mathbf{Q}_t \mathbf{Q}_t^\top \hat{\nabla}_{\mathbf{W}} f\left(\mathbf{W}_t; \xi_t\right) \mathbf{R}_t \mathbf{R}_t^\top \\[2mm]
\mathbf{G}_t^{\text{SAM}} = \nabla_{\mathbf{W}} f\left(\mathbf{W}_t + \rho \dfrac{\mathbf{G}_t^{\text{Pert}}}{\left\|\mathbf{G}_t^{\text{Pert}}\right\|_F}; \xi_t\right) \\[4mm]
\hat{\mathbf{G}}_t = \mathbf{Q}_t \rho_t \left(\mathbf{Q}_t^\top \mathbf{G}_t^{\text{SAM}} \mathbf{R}_t\right) \mathbf{R}_t^\top \\[2mm]
\mathbf{W}_T = \mathbf{W}_0 + \eta \displaystyle\sum_{t=0}^{T-1} \hat{\mathbf{G}}_t,
\end{cases}
$$

where $\rho_t$ is an entry-wise stateful gradient regularizer (e.g., Adam), $\mathbf{Q}_t \in \mathbb{R}^{m \times r}$ and $\mathbf{R}_t \in \mathbb{R}^{r \times n}$ are projection matrices, $T \in \mathbb{N}$ is the subspace update time, $\eta$ is the learning rate, and $\xi_t$ is a stochastic batch.

Note that Definition A.1 describes the Lorenza step in a general form. However, the perturbation projection is actually implemented more efficiently (as in Algorithm 2).

Following, we present the Algorithm 3 Truncated Randomized SVD, which efficiently approximates the leading singular vectors and values of a large matrix while reducing computational cost compared to the classical SVD.

---

**Algorithm 3** Truncated Randomized SVD

---

**Input:** Matrix $\mathbf{A} \in \mathbb{R}^{m \times n}$, target rank $r$, oversampling $p$, power iter. $q$
Draw $\mathbf{\Omega} \in \mathbb{R}^{n \times (r+p)}$, set $\mathbf{Y} = \mathbf{A}\mathbf{\Omega}$
**for** $i = 1$ to $q$ **do**
   $\mathbf{Y} = \mathbf{A}(\mathbf{A}^\top \mathbf{Y})$
**end for**
$\mathbf{Q} = \text{orth}(\mathbf{Y})$
$\mathbf{B} = \mathbf{Q}^\top \mathbf{A}$, then $\mathbf{B} = \tilde{\mathbf{U}} \mathbf{\Sigma} \mathbf{V}^\top$
$\mathbf{U} = \mathbf{Q}\tilde{\mathbf{U}}$
**Return:** Truncated $\mathbf{U}_r, \mathbf{V}_r$

---

The computational efficiency of this method stems from its key operations: multiplying the matrix $\mathbf{A}$ with a random projection matrix $\mathbf{\Omega}$ (complexity $O(mnr)$) and performing a QR decomposition on the resulting matrix $\mathbf{Y}$ (complexity $O(mr^2)$). The overall complexity, $O(mnr + mr^2)$, simplifies to $O(mnr)$ when $r \ll m, n$, making it significantly more scalable than the exact SVD, which requires $O(\min(mn^2, m^2n))$. This approach enables efficient extraction of leading singular vectors in large-scale data, making it well-suited for computationally constrained scenarios.

# B Proofs

## B.1 Proof Theorem 3.1

*Consider a $\beta$-smooth, non-convex function $f$ parametrized by a matrix $\mathbf{W} \in \mathbb{R}^{m \times n}$, where $m \leq n$, without loss of generality. Suppose $f$ satisfying $\sup_{\mathbf{W}} \mathbb{E}_\xi \|f(\mathbf{W}; \xi)\| \leq C$ for some large $C \in \mathbb{R}_+$ then, Algorithm 1 initialized at $\mathbf{W}_0$ with step size $\eta = \frac{1}{\beta\sqrt{T}}$,*

$$
\frac{1}{T} \sum_{t=0}^{T-1} \mathbb{E} \left\|\hat{\nabla} f\left(\mathbf{W}_t\right)\right\|_F^2 \leq \mathcal{O}\left(\frac{C\beta}{\sqrt{T}}\right) + \beta^2 \rho^2.
$$

where $\hat{\nabla} f(\mathbf{W}_t)$ is the RGE (1) of function $f$ with $q = 1, \mu \to 0$, and $\xi \sim \mathbb{P}_{\mathcal{D}}$ is a stochastic batch, drawn by distribution $\mathbb{P}_{\mathcal{D}}$ over domain $\mathcal{D}$.

*Proof.* First, consider the following notation. For the simplicity of writing, we let $\nabla f(\mathbf{W}) = \nabla_{\mathbf{W}} f(\mathbf{W}; \xi)$, where $\xi \sim \mathbb{P}_{\mathcal{D}}$ is a stochastic input batch, and $\mathbb{P}_{\mathcal{D}}$ is the sampling distribution over dataset/domain $\mathcal{D}$. Accordingly, we denote $\mathbb{E}[\nabla f(\mathbf{W})] = \mathbb{E}_{\xi}[\nabla_{\mathbf{W}} f(\mathbf{W}; \xi)]$. We denote the estimated gradient at $\mathbf{W}$ by

$$\mathbb{E}_{\xi}\left[\hat{\nabla}_{\mathbf{W}} f(\mathbf{W}; \xi)\right] = \frac{1}{q} \sum_{i=1}^{q} \mathbb{E}_{\xi}\left[\frac{f(\mathbf{W} + \mu \mathbf{U}_i; \xi) - f(\mathbf{W} - \mu \mathbf{U}_i; \xi)}{2\mu} \mathbf{U}_i\right] \in \mathbb{R}^{m \times n},$$

where $\mathbf{U}_i^{m \times n} \sim \mathcal{N}(\mathbf{0}, 1/n)$ is a randomized matrix. Similarly, for simplicity of writing, we denote $\hat{\nabla} f(\mathbf{W}) = \mathbb{E}_{\xi}\left[\hat{\nabla}_{\mathbf{W}} f(\mathbf{W}; \xi)\right]$. Notice that as $\mu \to 0$ and $q = 1$, the finite difference of the function values in approaches $f'(\mathbf{W}_t, \mathbf{U}_i) := \mathrm{Tr}\left(\nabla f(\mathbf{W}_t)^{\top} \mathbf{U}_i\right)$, denoting the directional derivative of $f(\mathbf{W})$, along the random direction $\mathbf{U}_i$, yielding $\hat{\nabla} f(\mathbf{W}_t) \to f'(\mathbf{W}_t, \mathbf{U}_i) \mathbf{U}_i$, thus,

$$\lim_{\mu \to 0} \mathbb{E}\left\|\hat{\nabla} f(\mathbf{W}_t)\right\|_F^2 = \mathbb{E}_{\mathbf{U}}\left\|f'(\mathbf{W}_t) \mathbf{U}_i\right\|_F^2 = \mathbb{E}_{\mathbf{U}}\left\|\mathrm{Tr}\left(\nabla f(\mathbf{W}_t)^{\top} \mathbf{U}_i\right) \mathbf{U}_i\right\|_F^2 = \mathbb{E}\left\|\nabla f(\mathbf{W}_t)\right\|_F^2. \tag{4}$$

For the simplicity of writing, let $\mathbf{X}_t = \mathbf{W}_t + \rho \frac{\hat{\nabla} f(\mathbf{W}_t)}{\|\hat{\nabla} f(\mathbf{W}_t)\|}$, thus $\mathbf{W}_{t+1} = \mathbf{W}_t - \eta \nabla f(\mathbf{X}_t)$. Now, by the $\beta$-smoothness of $f$, we have

$$\begin{aligned}
\mathbb{E} f(\mathbf{W}_{t+1}) \leq & \mathbb{E} f(\mathbf{W}_t) + \mathbb{E}\left[\mathrm{vec}(\nabla f(\mathbf{W}_t))^{\top} \mathrm{vec}(\mathbf{W}_{t+1} - \mathbf{W}_t)\right] + \frac{\beta}{2} \mathbb{E}\|\mathbf{W}_{t+1} - \mathbf{W}_t\|_F^2 \\
\underset{(I)}{=} & \mathbb{E} f(\mathbf{W}_t) - \eta \mathbb{E}\left[\mathrm{vec}(\nabla f(\mathbf{W}_t))^{\top} \mathrm{vec}(\nabla f(\mathbf{X}_t))\right] + \frac{\beta \eta^2}{2} \mathbb{E}\|\nabla f(\mathbf{X}_t)\|_F^2 \\
= & \mathbb{E} f(\mathbf{W}_t) - \frac{\eta}{2} \mathbb{E}\|\nabla f(\mathbf{W}_t)\|_F^2 - \frac{\eta}{2} \mathbb{E}\|\nabla f(\mathbf{X}_t)\|_F^2 + \frac{\eta}{2} \mathbb{E}\|\nabla f(\mathbf{W}_t) - \nabla f(\mathbf{X}_t)\|_F^2 \\
& + \frac{\beta \eta^2}{2} \mathbb{E}\|\nabla f(\mathbf{X}_t)\|_F^2 \\
\underset{(II)}{\leq} & \mathbb{E} f(\mathbf{W}_t) - \frac{\eta}{2} \mathbb{E}\|\nabla f(\mathbf{W}_t)\|_F^2 + \frac{\beta^2 \eta}{2} \mathbb{E}\|\mathbf{W}_t - \mathbf{X}_t\|_F^2 \\
\underset{(III)}{=} & \mathbb{E} f(\mathbf{W}_t) - \frac{\eta}{2} \mathbb{E}\|\nabla f(\mathbf{W}_t)\|_F^2 + \frac{\beta^2 \rho^2 \eta}{2} \\
\underset{(IV)}{=} & \mathbb{E} f(\mathbf{W}_t) - \frac{\eta}{2} \mathbb{E}\left\|\hat{\nabla} f(\mathbf{W}_t)\right\|_F^2 + \frac{\beta^2 \rho^2 \eta}{2},
\end{aligned} \tag{5}$$

where $(I)$ follows by the definition of $\mathbf{X}_t$, $(II)$ follows from $\frac{\eta}{2} = \frac{1}{2\beta\sqrt{T}} \geq \frac{1}{2\beta T} = \frac{\beta \eta^2}{2}$, $(III)$ follows by $\|\mathbf{X}_t - \mathbf{W}_t\|_F = \left\|\rho \frac{\hat{\nabla} f(\mathbf{W}_t)}{\|\hat{\nabla} f(\mathbf{W}_t)\|}\right\|_F = \rho$, and finally $(IV)$ follows by Equation (4). Rearrearage both sides to bound the gradient Forbinus norm, we obtain

$$\mathbb{E}\left\|\hat{\nabla} f(\mathbf{W}_t)\right\|_F^2 \leq \frac{2}{\eta}\left(\mathbb{E} f(\mathbf{W}_t) - \mathbb{E} f(\mathbf{W}_{t+1})\right) + \beta^2 \rho^2.$$

Adding up the inequality for $t \in [T-1]$, and dividing both sides by $T$, we get

$$\frac{1}{T} \sum_{t=0}^{T-1} \mathbb{E}\left\|\hat{\nabla} f(\mathbf{W}_t)\right\|_F^2 \leq \frac{2}{\eta T}\left(\mathbb{E} f(\mathbf{W}_0) - \mathbb{E} f(\mathbf{W}_T)\right) + \beta^2 \rho^2$$

$$\leq \frac{2C}{\eta T} + \beta^2 \rho^2$$

Now, choosing $\eta = \frac{1}{\beta\sqrt{T}}$, we have

$$\frac{2C}{\eta T} + \beta^2 \rho^2 \leq \frac{2C\beta}{\sqrt{T}} + \beta^2 \rho^2,$$

thus

$$\frac{1}{T} \sum_{t=0}^{T-1} \mathbb{E} \left\| \hat{\nabla} f \left( \mathbf{W}_t \right) \right\|_F^2 \leq \mathcal{O} \left( \frac{C\beta}{\sqrt{T}} \right) + \beta^2 \rho^2.$$

$\square$

## B.2  Proof of Theorem 3.2

*Consider a $\beta$-smooth nonconvex composition of $f \equiv \mathcal{L}\left(\Phi(\cdot)\right)$ that is bounded by some $M \in \mathbb{R}_+$. Let $\mathbf{G}_t^j$ denote the gradient matrix w.r.t. the $j$-th reversible layer $\mathbf{W}_t^j$, at time $t \in \mathbb{N}$, for all $j \in [L]$ and $t \in \mathbb{N}$, and $\mathsf{T}_\ell, \ell \in \mathbb{N}$ times are set by a convergence criterion (that is, $\|\hat{\mathbf{G}}_{\mathsf{T}_\ell}\| \leq \varsigma_\ell$). Consider any decay perturbation $\rho$ then, for any $\varepsilon > 0$, there exist $\mathsf{C} \in \mathbb{R}_+$ and $N$ such that for all $\mathsf{T}_N > \frac{\mathsf{C}}{\varepsilon^2}$, $\frac{1}{\mathsf{T}_N} \sum_{i=0}^{N-1} \sum_{t=\mathsf{T}_i}^{\mathsf{T}_{i+1}-1} \left\| \mathbf{G}_t^{j\,SAM} \right\|_F^2 \leq \varepsilon$.*

*Principally, Algorithm 2, with vanilla SGD weight update[2], achieves an $\varepsilon$-critical point,[3] i.e., $\left\| \mathbf{G}_t^j \right\|_F^2 \leq \varepsilon$, for some $t \in \mathbb{N}$, and any $j \in [L]$.*

*Proof.* We denote by $\mathsf{T}_\ell \in \mathbb{N}$ the time index $t$ at which we update the subspace, at Block 1 of the algorithm, for the $\ell$-th time, for $\ell \in \mathbb{N}$. For the simplicity of writing, for the $j$-th layer $\mathbf{W}_j$, we omit $j$ from $\mathbf{W}^j$, and use instead $\mathbf{G}_t^j = \nabla_{\mathbf{W}^j} f\left(\boldsymbol{\theta}_t\right) = \nabla f\left(\mathbf{W}_t\right)$. In addition, for simplicity, we let $\mathbf{X}_t = \mathbf{W}_t + \rho \frac{\nabla f(\mathbf{W}_t)}{\|\nabla f(\mathbf{W}_t)\|}$, thus $\mathbf{W}_{t+1} = \mathbf{W}_t - \eta \nabla f\left(\mathbf{X}_t\right)$. Consider the SVD decomposition of the gradient $\nabla_{\mathbf{W}^j} f\left(\boldsymbol{\theta}_{\mathsf{T}_i}\right) = \mathbf{U}_{\mathsf{T}_i} \Sigma_{\mathsf{T}_i} \mathbf{V}_{\mathsf{T}_i}^\top$. Accordingly, for $t \in [\mathsf{T}_i, \mathsf{T}_{i+1} - 1]$, we define the low rank gradient as $\hat{\mathbf{G}}_t \triangleq \mathbf{Q}_{\mathsf{T}_i} \mathbf{G}_t$, for $\mathbf{Q}_{\mathsf{T}_i} = \mathbf{U}_{\mathsf{T}_i}[:, :r] \mathbf{U}_{\mathsf{T}_i}[:, :r]^\top$, where $\mathbf{U}_{\mathsf{T}_i}$ is obtained by the subspace search, using the exact truncated SDV calculation. Now, let $h_t \triangleq \mathbb{E}f\left(\mathbf{W}_t\right) - \mathbb{E}f\left(\mathbf{W}_{\mathsf{T}_{i+1}}\right)$, and $\eta \equiv \eta_t$ denote the learning rate. Then,

$$
\begin{aligned}
h_{t+1} &= \mathbb{E}f\left(\mathbf{W}_{t+1}\right) - \mathbb{E}f\left(\mathbf{W}_{\mathsf{T}_{i+1}}\right) \\
&\underset{(I)}{\leq} \mathbb{E}f\left(\mathbf{W}_t\right) - \mathbb{E}f\left(\mathbf{W}_{\mathsf{T}_{i+1}}\right) - \frac{\eta}{2}\mathbb{E}\|\nabla f\left(\mathbf{W}_t\right)\|_F^2 + \frac{\beta^2 \rho^2 \eta}{2} \\
&\underset{(II)}{=} h_t - \frac{\eta}{2}\mathbb{E}\left\|\hat{\nabla} f\left(\mathbf{W}_t\right)\right\|_F^2 + \frac{\beta^2 \rho^2 \eta}{2},
\end{aligned}
\tag{6}
$$

where $(I)$, follows Equation (5), and $(II)$ follows by Equation (4). Rearranging equation 6, and choosing $\eta_t = \eta$, for all $t \geq 0$, we readily obtain that,

$$\sum_{t=\mathsf{T}_i}^{\mathsf{T}_{i+1}-1} \mathbb{E}\left\|\hat{\nabla} f\left(\mathbf{W}_t\right)\right\|_F^2 \leq \frac{2(h_{\mathsf{T}_i} - h_{\mathsf{T}_{i+1}})}{\eta} + (\mathsf{T}_{i+1} - \mathsf{T}_i)\beta^2 \rho^2.$$

Thus, for $N \in \mathbb{N}$,

$$
\begin{aligned}
\frac{1}{\mathsf{T}_N} \sum_{i=0}^{N-1} \sum_{t=\mathsf{T}_i}^{\mathsf{T}_{i+1}-1} \mathbb{E}\left\|\hat{\nabla} f\left(\mathbf{W}_t\right)\right\|_F^2 &\leq \frac{1}{\mathsf{T}_N} \sum_{i=0}^{N-1} \left[ \frac{2(h_{\mathsf{T}_i} - h_{\mathsf{T}_{i+1}})}{\eta} + (\mathsf{T}_{i+1} - \mathsf{T}_i)\beta^2 \rho^2 \right] \\
&= \frac{2(h_{\mathsf{T}_0} - h_{\mathsf{T}_N})}{\eta \mathsf{T}_N} + \frac{(\mathsf{T}_N - \mathsf{T}_0)\beta^2 \rho^2}{\mathsf{T}_N} \\
&\leq \frac{M}{\beta \sqrt{\mathsf{T}_N}} + \beta^2 \rho^2,
\end{aligned}
\tag{7}
$$

where $\eta = \frac{1}{\beta \sqrt{\mathsf{T}_N}}$. Now by the definition of $\mathbf{Q}_{\mathsf{T}_i}$ for any $i \in \mathbb{N}$ there exists some $\alpha \in (0, 1]$, for which

$$\left\| \hat{\nabla} f\left(\mathbf{W}_{\mathsf{T}_i}\right) - \mathbf{Q}_{\mathsf{T}_i} \hat{\nabla} f\left(\mathbf{W}_{\mathsf{T}_i}\right) \right\|_F^2 \leq \alpha \left\| \hat{\nabla} f\left(\mathbf{W}_{\mathsf{T}_i}\right) \right\|_F^2.
\tag{8}$$

---

[2]We focus on SGD for simplicity (as is standard practice in related literature, e.g., (Zhao et al., 2024)).

[3]Also known as $\varepsilon$-stationary, see, e.g., (Cosson et al., 2023).

Obviously, the following clearly holds for any $t \in \mathbb{N}$,

$$
\begin{aligned}
\left\|\hat{\nabla} f\left(\mathbf{W}_t\right)\right\|_F^2 &= \left\|\mathbf{Q}_{\mathsf{T}_i} \hat{\nabla} f\left(\mathbf{W}_t\right)\right\|_F^2 + \left\|\mathbf{Q}_{\mathsf{T}_i}^{\perp} \hat{\nabla} f\left(\mathbf{W}_t\right)\right\|_F^2 \\
&= \left\|\mathbf{Q}_{\mathsf{T}_i} \hat{\nabla} f\left(\mathbf{W}_t\right)\right\|_F^2 + \left\|\hat{\nabla} f\left(\mathbf{W}_t\right) - \mathbf{Q}_{\mathsf{T}_i} \hat{\nabla} f\left(\mathbf{W}_t\right)\right\|_F^2,
\end{aligned}
\tag{9}
$$

and thus by plugging equation 8 into equation 9, at $t = \mathsf{T}_i$, for any $i \in \mathbb{N}$, we get, $(1 - \alpha)\|\hat{\nabla} f\left(\mathbf{W}_{\mathsf{T}_i}\right)\|_F^2 \leq \left\|\mathbf{Q}_{\mathsf{T}_i} \hat{\nabla} f\left(\mathbf{W}_{\mathsf{T}_i}\right)\right\|_F^2$. Accordingly,

$$
\left\|\mathbf{Q}_{\mathsf{T}_i}^{\perp} \hat{\nabla} f\left(\mathbf{W}_{\mathsf{T}_i}\right)\right\|_F^2 \leq \frac{\alpha}{1 - \alpha} \left\|\mathbf{Q}_{\mathsf{T}_i} \hat{\nabla} f\left(\mathbf{W}_{\mathsf{T}_i}\right)\right\|_F^2.
\tag{10}
$$

Recall from Lemma B.3, Equation 31, in (Zhao et al., 2024) that for the reversible layer,

$$
\mathbb{E}\left\|\hat{\nabla} f\left(\mathbf{W}_t\right)\right\|_F^2 = \mathbb{E}\left\|(I - \eta \mathbf{S})\hat{\nabla} f\left(\mathbf{W}_{t-1}\right)\right\|_F^2
\tag{11}
$$

$$
\leq \|(I - \eta \mathbf{S})\|_2^2 \, \mathbb{E}\left\|\hat{\nabla} f\left(\mathbf{W}_{t-1}\right)\right\|_F^2
\tag{12}
$$

$$
= \max_i |1 - \eta \lambda_i|^2 \mathbb{E}\left\|\hat{\nabla} f\left(\mathbf{W}_{t-1}\right)\right\|_F^2,
\tag{13}
$$

where $\{\lambda_i\}_i$ are the eigenvalue of $\mathbf{S}$. Thus, using the fact that $\mathbf{S}$ is positive semi-definite matrix, If the learning rate $\eta$ satisfies $\eta \leq \frac{2}{\lambda_{\max}}$, where $\lambda_{\max}$ is the maximum eigenvalue of $\mathbf{S}$, it follows that $\max_i |1 - \eta \lambda_i|^2 \leq 1$. Consequently, this implies that $\mathbb{E}\left\|\hat{\nabla} f\left(\mathbf{W}_t\right)\right\|_F^2 \leq \mathbb{E}\left\|\hat{\nabla} f\left(\mathbf{W}_{t-1}\right)\right\|_F^2$. Therefore, the Frobenius norm of the gradient decreases monotonically as a function of time $t$. Now, recall that $\varsigma_i$ is any positive number such that $\varsigma_i < \sqrt{1 - \alpha} \cdot \|\hat{\nabla} f\left(\mathbf{W}_{\mathsf{T}_{i-1}}\right)\|_F$. According to (10), this necessarily implies that in each block $i$, we will execute (at least once) the low-rank optimization block (indeed, the condition $\|\hat{\nabla} f\left(\mathbf{W}_t\right)_{\mathsf{T}_i}\|_F > \varsigma_i$ is satisfied). This, conjugated with the monotonicity property that $\left\|\hat{\nabla} f\left(\mathbf{W}_t\right)\right\|_F^2 \leq \left\|\hat{\nabla} f\left(\mathbf{W}_{\mathsf{T}_i}\right)\right\|_F^2$, for any $t \in [\mathsf{T}_i, \mathsf{T}_{i+1} - 1]$ and $i \in [N]$, imply that

$$
\frac{1}{\mathsf{T}_N} \sum_{i=0}^{N-1} \sum_{t=\mathsf{T}_i}^{\mathsf{T}_{i+1}-1} \left\|\hat{\nabla} f\left(\mathbf{W}_t\right)\right\|_F^2 \leq \frac{1}{\mathsf{T}_N} \sum_{i=0}^{N-1} \sum_{t=\mathsf{T}_i}^{\mathsf{T}_{i+1}-1} \left\|\hat{\nabla} f\left(\mathbf{W}_{\mathsf{T}_i}\right)\right\|_F^2
\tag{14}
$$

$$
\leq \frac{1}{(1-\alpha)\mathsf{T}_N} \sum_{i=1}^{N-1} \sum_{t=\mathsf{T}_i}^{\mathsf{T}_{i+1}-1} \left\|\mathbf{Q}_{\mathsf{T}_i} \hat{\nabla} f\left(\mathbf{W}_t\right)\right\|_F^2
\tag{15}
$$

$$
\leq \frac{M}{(1-\alpha)\beta\sqrt{\mathsf{T}_N}} + \frac{\beta^2 \rho^2}{1 - \alpha}.
\tag{16}
$$

Accordingly, for decaying perturbation $\rho \equiv \rho_t$, without the loss of generality for any small enough $\varepsilon_1, \varepsilon_2 \geq 0$, there exist $\varepsilon \geq \varepsilon_1 + \varepsilon_2$, where $\mathsf{T}_N > \frac{M^2}{(1-\alpha)^2 \varepsilon_1^2} \geq \frac{\mathsf{C}}{\varepsilon^2}$, and $\rho_N < \varepsilon_2 \frac{1-\alpha}{\beta^2}$,

$$
\min_{0 \leq t \leq \mathsf{T}_N} \left\|\hat{\nabla} f\left(\mathbf{W}_t\right)\right\|_F^2 \leq \frac{1}{\mathsf{T}_N} \sum_{i=0}^{N-1} \sum_{t=\mathsf{T}_i}^{\mathsf{T}_{i+1}-1} \left\|\hat{\nabla} f\left(\mathbf{W}_t\right)\right\|_F^2 \leq \varepsilon_1 + \varepsilon_2 \leq \varepsilon
$$

and thus, there exists an iteration index $t \in [0, \mathsf{T}_N]$ for which,

$$
\left\|\hat{\nabla} f\left(\mathbf{W}_t\right)\right\|_F^2 \leq \varepsilon,
\tag{17}
$$

which, by definition, implies that Algorithm 2 achieves an $\varepsilon$-critical point.

$\square$

## C  Additional definitions

**Definition C.1.** (Reversibility (Tian et al., 2021))  A neural network $\phi$ that maps input $\boldsymbol{x}$ to output $\boldsymbol{y} = \phi(\boldsymbol{x}; \theta)$ is reversible, if there exists $L(\boldsymbol{x}; \theta)$ so that $\boldsymbol{y} = L(\boldsymbol{x}; \theta)\boldsymbol{x}$, and the backpropagated gradient $\boldsymbol{g_x}$ satisfies $\boldsymbol{g_x} = L^{\top}(\boldsymbol{x}; \theta)\boldsymbol{g_y}$, where $\boldsymbol{g_y}$ is the backpropagated gradient at the output $\boldsymbol{y}$. $L(\boldsymbol{x}; \theta)$ depends on the input $\boldsymbol{x}$ and weight $\theta$ in the network $\phi$.

## D  Evaluation on Zero-Shot Reasoning Tasks - additional details

Specifically, we evaluate the pretrained models on the following tasks:

- **Perplexity:** Measured on the C4 dataset (Raffel et al., 2020).

- **Commonsense Reasoning:** BoolQ (Clark et al., 2019), RTE (Wang et al., 2018), HellaSwag (HS) (Zellers et al., 2019), Winogrande (WG) (Sakaguchi et al., 2021), OpenBookQA (OBQA) (Mihaylov et al., 2018), ARC-Easy (ARC-E), and ARC-Challenge (ARC-C) (Clark & Etzioni, 2018).

- **Physical and Scientific Reasoning:** PIQA (Bisk et al., 2020), SciQ (Johannes Welbl, 2017), and MathQA (Amini et al., 2019).

To ensure a fair comparison, we repeat the experimental setup used for AdamW, APOLLO, and APOLLO-Mini. The results are summarized in the table below.

