# OpenReview forum: "Lorenza: Enhancing Generalization in Low-Rank Gradient LLM Training via Efficient Zeroth-Order Adaptive SAM"
_TMLR — Accepted by TMLR_

### Review · Reviewer_De7q · 2025-10-29

**Summary Of Contributions:**

The paper introduces Lorenza, which is a memory-efficient optimizer that combines Sharpness-Aware Minimization (SAM) with low-rank gradient updates to improve generalization in LLM training. The authors provide theoretical foundations for the method. The method is evaluated in pre-training, pos-training and zero-shot reasoning benchmarks.

**Audience:**

Yes

**Audience Explanation:**

This work focus on the main part (optimizer) of language model training.

**Claims And Evidence:**

Yes

**Claims Explanation:**

I'm not an expert of this field, but the theoretical part could set a good foundation of the method. However, my concern is regarding the assumption of the method could be too strict and hard to generalize.

**Requested Changes:**

1. The experimental results although beat the baselines, the improvement is marginal. Shows 0.5-2% gains over baselines across GLUE, pre-training, and reasoning tasks - improvements that, while statistically questionable due to missing confidence intervals. Lorenza+ shows better results, but it is essentially Lorenza + standard gradient component.

2. The reversible network assumption is so limited, and could only applies to specific architectures. This is also the foundation of the while 3.2 theorem.

3. I'm also worried about the zeroth-order approximation. Numerical precision limits how small μ can be. And there is no analysis of finite μ effects on convergence. This also make me confuse about why it works so well.

---

> ### Author Response · Authors · 2025-12-09
>
> We thank the reviewer for the constructive feedback and for highlighting the method's theoretical foundation. We also acknowledge the valuable concerns raised and address them in the following response.
>
> **Clarification regarding the assumptions:** We note that our theoretical assumptions are weaker than those commonly used in prior top-tier work on low-rank optimization for LLM training, which allows us to analyze a richer setting. For instance, to the best of our knowledge, the convergence analysis in all existing low-rank training methods (scoped in the introduction) focuses on reversible layers and, unlike our approach, treats the projection subspace as fixed throughout training, thus omitting subspace update dynamics. We did not rely on the simplifying assumption of a fixed subspace, as assumed before our analysis, but instead aim to advance convergence analysis for evolving subspaces. We also observe that many related papers (E.g., Apollo, Adam-mini) that report competitive or lower empirical performance do not include convergence theory for their optimization methods. We hope this context clarifies how our analysis complements and extends prior art.
>
> **Q1. GLUE gains and variance:**
> We appreciate the reviewer’s concern regarding the magnitude of improvements on GLUE. In this area, gains from new optimization methods are often very tight, and we aim to report them transparently. We have therefore added measurement variance to the GLUE table to better convey stability and effect size.
> Additionally, Lorenza+ introduces a new element for low-dimensional optimization: at each step, it incorporates the orthogonal component of the gradient when updating within the subspace. To the best of our knowledge, this mechanism has not appeared before in this setting and helps explain the consistent improvements we observe. It does not require any additional memory, and this contribution can be helpful to and incorporated in all subspace-based optimizers. The intuition behind Lorenza+ can be summarized as "Low-Rank Adaptation with Full-Rank Compensation": Standard low-rank optimizers (like GaLore or vanilla Lorenza) project the gradient into a small subspace and discard the rest. This completely ignores gradient components orthogonal to that subspace ($G^{SAM \perp}$), which may still contain valuable information for learning and is already calculated!. Our suggested solution (Lorenza+) is to utilize this "lost" information.
>
> **Q2. Assumptions in Theorems 3.2 and 3.1:**
> We would like to recall, as mentioned above, that our assumptions are weaker than suggested in previous works, and capture the subspace update dynamics that were omitted in previous analysis. The other reviewer found our convergence analysis to be of broad interest to researchers interested in fast optimization of LLMs. Please note that the definition of "reversible layers" (or layers exhibiting the requisite spectral properties) encompasses linear layers (MLPs and convolutional layers) and ReLUs. These components constitute the bulk of the Transformer architectures (RoBERTa, LLaMA) used in our experiments, justifying the assumption.
>
> For Theorem 3.1, the convergence of Algorithm 1 (AdaZo-SAM), no reversibility assumption is required. The proof is architecture-agnostic and applies generally. We made this explicit in the revised manuscript to avoid any ambiguity.
> As for the bounded-loss assumption ($\sup \mathbb{E}[f] \leq C$) is a standard analytical tool in non-convex optimization proofs to ensure the summation of gradients is finite (used for example in LDAdam, Galore, Apollo, AdaRankGrad, SUMO). While the Cross-Entropy (CE) loss is theoretically unbounded as probabilities approach zero, in practical LLM training, the loss remains effectively bounded because gradient-driven optimization ensures non-zero probabilities for target tokens immediately after initialization. This makes the bounded-loss assumption a realistic approximation of the actual training dynamics rather than a theoretical violation.
>
> **Q3. Choice of μ:**
> We appreciate the question about μ. Intuitively, selecting a small μ injects mild noise, akin to the inherent stochasticity of SGD, which can be beneficial for learning in practice. We treat μ as a hyperparameter; empirically, choosing it sufficiently small (as noted in the paper) worked well across all experiments. This approach worked well in practice for other methods, such as [1,2], which were shown to be effective for replacing backpropagation in low-resource settings.
>
>
> [1] Fine-Tuning Language Models with Just Forward Passes. Yihua Zhang et al. NeurIPS 2023.
>
> [2] Revisiting Zeroth-Order Optimization for Memory-Efficient LLM Fine-Tuning: A Benchmark. Yihua Zhang et al.

---

### Review · Reviewer_BZsz · 2025-11-19

**Summary Of Contributions:**

The paper proposes a family of memory-efficient sharpness-aware optimizers for training large neural networks based on zeroth-order perturbation estimation and low-rank gradient projection.

The first contribution in Section 3.1 is an optimizer AdaZo-SAM  that imitates SAM’s two-step update but avoids the expensive second gradient computation. The key idea is to use a zeroth-order finite-difference estimator with a single random probe to reduce the cost in back propagation. Theorem 3.1 provides  convergence analysis.

The second contribution is Lorenza, which extends AdaZo-SAM by projecting gradients into a low-rank subspace using randomized SVD. This reduces memory footprint while still attempting to capture high-curvature directions within a subspace. Theorem 3.2 provides  conference analysis.

The third contribution extends Lorenza with an additional correction term that reintroduces the gradient component outside the low-rank subspace at essentially no additional memory cost. This is intended to reduce the generalization gap caused by low-rank projection.

**Additional Comments:**

Please address the questions above.

**Audience:**

Yes

**Audience Explanation:**

I believe the convergence analysis would be of broad interest to researchers interested in fast optimization of LLMs.

**Broader Impact Concerns:**

Ethical implications do not seem applicable.

**Claims And Evidence:**

No

**Claims Explanation:**

Although the paper claims to implement sharpness aware minimization (SAM), I do not see this supported in the actual theoretical results. Theorem 3.1 and 3.2 do not provide any guarantees that is consistent with SAM. They only provide standard convergence results.

Although the algorithms are somewhat inspired by SAM as far as I can see the proposed approach does not approximate the SAM inner maximization nor optimize the SAM minimax objective.

**Requested Changes:**

Please address the following questions:

Theorem 3.1:

1. How do your results hold for realistic $\mu$ rather than $\mu \rightarrow 0$?
2. Why does the variance term that scales as the dimension $d$ not appear in your bound?
3. The theorem does not analyze the SAM objective at all. How does it ensure SAM-like behavior?
4. The bounded-loss assumption is violated for CE-based LLM losses. Can you clarify this discrepancy?

Theorem 3.2
1. The theorem assumes reversible layers. Is this consistent with the models used in the experiments?
2. There is no bound on the R-SVD approximation error. How does this affect convergence?
3. How does the analysis incorporate the noise introduced by the q=1 zeroth-order ascent?

---

> ### Author Response · Authors · 2025-12-09
>
> We thank the reviewer for their rigorous examination of our theoretical results and for noting that the convergence analysis would be of broad interest to researchers seeking to optimize LLMs quickly. We appreciate the opportunity to clarify the relationship between our convergence analysis and the SAM approach, as well as the specific mechanics of the Zeroth-Order (ZO) estimator.
>
> **Clarification regarding the accuracy of our claims:** We respectfully clarify that Lorenza does not perform Zeroth-Order Descent. Instead, we employ a Zeroth-Order estimator strictly to approximate the ascent direction (the perturbation), while the descent step (the weight update) remains a First-Order operation using analytical gradients.
> Regarding the theoretical objective, our analysis is not merely a 'standard convergence result' but explicitly models the Min-Max dynamics of SAM, accounting for both the inner ascent step and the outer descent step. A clear indicator of this is the convergence rate itself: our bound in Theorem 3.1 includes a constant term that depends on the radius $\rho$ but is independent of $T$. This term is the signature of SAM convergence theory (distinct from standard SGD) and represents the 'flatness' regularization floor. Thus, we demonstrate that the noise introduced by the randomized ascent direction ($q=1$) does not break the unique convergence guarantees of the SAM framework.
> Theorem 3.2 extends this analysis to the dynamic low-rank setting. It proves convergence to a stationary point even when the optimization subspace evolves during training, and the ascent perturbation is estimated via the Zeroth-Order estimator within that subspace.

---

> ### Author Response · Authors · 2025-12-09
> **Theorem 3.1**
>
> **Q1. realistic $\mu$:** The assumption $\mu \to 0$ allows us to utilize the directional derivative limit for a clean theoretical derivation. For realistic, non-zero (but small) $\mu$, the estimator becomes a finite-difference approximation. Standard results in ZO optimization (e.g., Nesterov & Spokoiny, 2017) show that the error introduced by a finite $\mu$ scales with $\mu^2$ and the smoothness of $f$. Since Lorenza uses this estimator only for the perturbation direction (which is subsequently normalized), the impact of finite $\mu$ is further mitigated. The direction remains correlated with the ascent direction, which is sufficient for inducing the "flatness" regularizing effect, even if the magnitude is slightly inexact.
>
> **Q2. $d$-dependent variance:**
> This is a critical distinction arising from our hybrid framework. The $d$-dependent variance penalty typically appears in pure Zeroth-Order Optimization, where the descent step is estimated. In Lorenza (Algorithm 1), the update rule is $W_{t+1} = W_t - \eta \nabla_W f(W_t + \epsilon)$. The gradient $\nabla_W f$ used for the parameter update is an analytical First-Order gradient, not a ZO estimate. The ZO estimator is used only to find $\epsilon$. Therefore, the weight update step does not suffer from a potential variance of a ZO estimator, and the convergence bound is dominated by the properties of the First-Order gradient, not the dimensional scaling of the perturbation estimator.
>
> **Q3. Consistency with SAM-like behavior:** We analyze convergence to the stationary points of $f(w)$, consistent with analyses in AdaSAM. The "SAM-like behavior" is structural rather than purely a consequence of the convergence bound. By stepping in the direction of the gradient evaluated at a perturbed point $w + \epsilon$, the optimizer inherently penalizes sharp minima. Our theorem shows that this perturbation mechanism, when approximated by our efficient ZO method, does not prevent convergence to a valid minimum.
>
> **Q4. bounded-loss assumption:** We acknowledge that Cross-Entropy is theoretically unbounded. However, the bounded-loss assumption ($\sup \mathbb{E}[f] \leq C$) is a standard analytical tool in non-convex optimization proofs to ensure the summation of gradients is finite (used for example in LDAdam, Galore, Apollo, AdaRankGrad, SUMO). While the Cross-Entropy (CE) loss is theoretically unbounded as probabilities approach zero, in practical LLM training, the loss remains effectively bounded because gradient-driven optimization ensures non-zero probabilities for target tokens immediately after initialization. This makes the bounded-loss assumption a realistic approximation of the actual training dynamics rather than a theoretical violation.

---

> ### Author Response · Authors · 2025-12-09
> **Theorem 3.2**
>
> Notably, our analysis is the first to model the full training dynamics of a time-varying projected subspace, distinct from prior works (scoped in the introduction) that rely on a simplified fixed-subspace assumption.
>
> **Q1. Reversible layers:** As noted in the paper, the definition of "reversible layers" (or layers exhibiting the requisite spectral properties) encompasses linear layers (MLP and convolutional) and ReLUs. These components constitute the bulk of the Transformer architectures (RoBERTa, LLaMA) used in our experiments, justifying the assumption. Note that this assumption is made in all suggested methods: Galore, AdaRankGrad, SUMO (ICML, ICLR, NeurIPS), etc.
>
> **Q2. R-SVD approximation error:** We prove the theorem for the exact SVD projection. In the revised version, we included this in the theorem and explicitly stated that this step is replaced by an approximation (R-SVD) to reduce complexity. We acknowledge this trade-off. While a full SVD identifies the optimal subspace for optimization, it incurs higher time and space complexity. In contrast, the R-SVD significantly reduces these computational costs with only a minimal impact on the convergence rate, as shown empirically in the ablation study section D.
>
> **Q3. Incorporation of th noise ($q=1$) in the analysis:** The noise is incorporated via the expectation operator $\mathbb{E}$. In Equation (3), in the proof, we show that as $\mu \to 0$, the expected squared norm of the estimator converges to the true gradient norm: $\mathbb{E} ||\hat{\nabla} f||^2 = \mathbb{E} ||\nabla f||^2$. This implies that while the individual ZO RGE (the ascent steps with $q=1$) are noisy, they are unbiased estimators of the true gradient expectation.

---

### Review · Reviewer_EzyA · 2025-11-29

**Summary Of Contributions:**

This paper introduces Lorenza (and its enhancement Lorenza+), a memory-efficient training algorithm that demonstrates good generalization performance, applicable to both pretraining and fine-tuning of large language models (LLMs). The algorithm combines zero-order gradient estimation, low-rank gradient updates, and Sharpness-Aware Minimization (SAM) to achieve memory savings while retaining the advantages of SAM in improving model generalization.

**Additional Comments:**

I recommend a **major revision** of the manuscript.

**Audience:**

Yes

**Audience Explanation:**

The proposed algorithms are of interest to researchers and practitioners focusing on memory-efficient LLM fine-tuning.

**Broader Impact Concerns:**

No ethical concerns

**Claims And Evidence:**

No

**Claims Explanation:**

1. Motivation is unclear: While the paper aims to propose a low-memory fine-tuning algorithm with generalization comparable to full fine-tuning, it does not clearly articulate why the sharpness-aware (flat-minima) principle of SAM is particularly beneficial in the fine-tuning context. Fine-tuning presents distinct challenge, such as domain shift, limited data, and catastrophic forgetting. The paper would benefit from a clearer discussion on how encouraging flat minima relates to these characteristics. Moreover, the pretraining experiments are also included, but pretraining faces its main memory bottleneck in activation values, not optimizer state, so the memory savings claimed by Lorenza might not be as crucial during pretraining.

2. Lack of intuitive understanding of the algorithm: The introduction of the correction term in Lorenza+ seems to be added directly to the gradient update, but the rationale behind this step is not explained intuitively. Is there a simpler or clearer explanation for why this correction term is necessary, and how it contributes to the performance improvement?

3. Insufficient experiments:
    - The fine-tuning experiments are only conducted on a Roberta-base model (with <1B parameters). It would be beneficial to include experiments on larger models, such as OPT (>1B), to better demonstrate the scalability of the method.
    - Furthermore, experimental settings and hyperparameters are not clearly stated. For instance, the frequency of performing SVD and other key details need to be included to ensure reproducibility.
    - From the experimental results, the advantage of Lorenza does not seem significant, particularly for pretraining tasks. The Lorenza+ variant shows more obvious improvements, which raises the question: Is the main contributing factor the correction term introduced in Lorenza+? More experiments and clarifications are needed here.

4. Additional concerns:
    - Algorithm details: In standard SAM, the normalization is performed using the 2-norm, but in Algorithm 1 and Algorithm 2, the normalization is done using the Frobenius norm. What is the reasoning behind this choice? In fact, the perturbation term , in the default case, is essentially just a scalar multiplication of a random normal matrix . This makes the perturbation direction essentially randomly determined with only the perturbation length differing. Why does this random choice of perturbation direction not lead to a significant degradation in performance, especially with its potentially high variance?
    - Inaccurate statements and improper citations: There are instances of imprecise statements and improper citation practices. For example, in the Introduction, it is stated that `GaLore reduces the number of fine-tuned parameters’, but GaLore mainly reduces the optimizer state size. Similarly, in Table 3, many of the experimental results appear to be derived from the paper of GaLore or Apollo, but these results should be cited properly and it should be confirmed that the numbers are consistent with the original papers.
    - Ablation experiments: The ablation studies are insufficient. It is recommended that these studies be presented in the main body of the paper rather than in the appendix. Additionally, ablation experiments should include results on how different components (e.g., correction term, low-rank updates, zero-order gradient) influence performance.

**Requested Changes:**

Major Changes

- Clarify Motivation and Connection to Fine-Tuning: The authors are encouraged rewrite the motivation to explicitly articulate why the sharpness-aware (flat-minima) principle is specifically beneficial for fine-tuning. The current text does not sufficiently connect SAM's objectives to unique fine-tuning challenges such as domain shift, limited data availability, or catastrophic forgetting.

- Provide Intuition for the "Correction Term": The introduction of the correction term in Lorenza+ currently lacks sufficient explanation. Please provide a clear intuitive or theoretical justification for why this term is added directly to the gradient update and how it specifically contributes to performance gains.

- Expand Experiments to Larger Models: The current evaluation on RoBERTa (<1B parameters) is insufficient to prove the method's utility for modern Large Language Models. The authors are requested to conduct fine-tuning experiments on models larger than 1B parameters (e.g., OPT-1.3B, Llama-series, or similar) to demonstrate the scalability and robustness of the proposed method.

- Strengthen Ablation Studies: (1) Placement: Key ablation studies should be moved from the Appendix to the main body of the paper. (2) Content: Expand the ablations to isolate the impact of specific components, particularly the correction term versus the low-rank updates. Given that Lorenza+ significantly outperforms Lorenza (which shows marginal gains), the authors must clarify if the performance improvement is primarily driven by the correction term rather than the core algorithm.

- Clarify Algorithm Mechanics: (1) Please explain the rationale for using the Frobenius norm for normalization in Algorithms 1 and 2, as opposed to the standard 2-norm used in vanilla SAM. (2) Provide a justification for the random perturbation strategy. Since the perturbation is essentially a random normal matrix scaled by a scalar, the direction is random. Please discuss why this high-variance approach does not lead to performance degradation.

Minor Changes.

- Improve Reproducibility: The authors are encouraged to explicitly list all experimental settings and hyperparameters, specifically details such as the frequency of SVD computation. A detailed hyperparameter table should be included to ensure the work is reproducible.

- Correct Inaccuracies and Citations:

    - Correct the statement regarding GaLore in the Introduction. GaLore primarily reduces optimizer state size, not the number of fine-tuned parameters.

    - In Table 3 (and similar tables), properly cite the sources of the baseline results (e.g., GaLore, Apollo). If numbers are taken directly from those papers, please clearly indicate this and verify consistency.

---

> ### Author Response · Authors · 2025-12-09
>
> We sincerely thank the reviewer for their constructive feedback. We value your insights regarding the motivation for sharpness-aware minimization in fine-tuning and the intuitive justification for Lorenza+. Below, we address your specific concerns.
>
> **Major concerns**
>
> **Motivation and Connection to Fine-Tuning.** We appreciate this opportunity to clarify this point. The motivation for using SAM in fine-tuning stems from challenges with distribution shift and robustness, rather than just in-domain accuracy; it is explicitly driven by limited (downstream) training data.
>
> OOD Generalization: As noted in our Introduction, standard memory-efficient optimizers often degrade performance on complex reasoning tasks (e.g., math, coding) that require robust generalization beyond the training set. Flat minima have been theoretically and empirically linked to better Out-Of-Distribution (OOD) generalization [1,2,3,4]. In many downstream fine-tuning scenarios, the training dataset is small and may not fully represent the Out-Of-Distribution (OOD) variance of the target task. Standard optimizers often converge to sharp minima that overfit the idiosyncrasies of these small datasets. The flat-minima principle used by Lorenza is particularly beneficial here because flat regions are theoretically shown to be less sensitive to parameter perturbations, leading to better invariance against distribution shifts. By encouraging flatness, Lorenza prevents the model from latching onto brittle features of a small fine-tuning set, thereby improving OOD generalization.
>
> Evidence: This is why we specifically included the GSM8K reasoning benchmark (Table 4). While Galore achieves 74.9% accuracy on LLaMA-1B (8-shot), our sharpness-aware approach, Lorenza, obtained 76.4% (Lorenza+ achieves 77.1%). This massive gap (+1.5%) validates the "flat-minima" principle as crucial for preserving LLMs' reasoning capabilities during fine-tuning, mitigating catastrophic forgetting or "dumbing down" often seen with low-rank adapters.
>
> **Intuitive Understanding of Lorenza+.** The intuition behind Lorenza+ can be summarized as "Low-Rank Adaptation with Full-Rank Compensation": Standard low-rank optimizers (like GaLore or vanilla Lorenza) project the gradient into a small subspace and discard the rest. This completely ignores gradient components orthogonal to that subspace ($G^{SAM \perp}$), which may still contain valuable information for learning and is already calculated!. Our suggested solution (Lorenza+) is to utilize this "lost" information. We split the update into two parts:
>
> The "Heavy" Lift: We use complex adaptive moments in SAM-style for the low-rank component (the most important directions).
>
> The "Light” Correction: We use a simple, memory-cheap SGD update for the orthogonal component (the discarded residuals).
>
> Note that we only take the orthogonal component of the gradient; thus, it does not contradict the step onto the low-rank subspace applied by the low-rank SAM optimization. Notably, it requires no additional memory and can be incorporated into all subspace-based optimizers.
>
> This hybrid approach allows us to update the model in all directions (effectively full rank) without the memory cost of storing full-rank optimizer states. As shown in Table 3, this correction recovers significant gains (e.g., reducing LLaMA-1B perplexity from 14.93 to 13.98) by leveraging gradient information that standard low-rank methods discard.
>
> **Experiments of Large Models.** We have conducted experiments with models exceeding 1B parameters, demonstrating the method's scalability.
> LLaMA-1B & Phi-2 (2.7B): In Table 4, we evaluate Lorenza on the GSM8K benchmark using Phi-2 (2.7B parameters) and LLaMA (1B parameters). Lorenza+ significantly outperforms GaLore and LoRA on these architectures.
> Pre-training Scalability: In Table 3, we scale our pre-training experiments up to LLaMA-1B on the C4 dataset, showing consistent gains over baselines.
> In the final revision, we highlight the 2.7B Phi-2 results more prominently in the main text to explicitly address the scalability concern. We will try to add a pre-training 7B-parameter experiment to Table 3 (since this experiment requires a long training time).
> Please note that previous SOTA accepted papers, such as Galore, Apollo, AdaRankGrad, SUMO (ICML, MlSys, ICLR, NeurIPS), etc., followed the same experimental standard, and our experimental section is no less extensive than theirs.

---

> ### Author Response · Authors · 2025-12-09
>
> **Algorithm Details.**
>
> Norms. Mathematically, for matrix parameters (which constitute the weights of neural networks), the Frobenius norm is mathematically equivalent to the vector $L_2$ norm of the flattened parameters. Practically, standard SAM implementations in PyTorch (e.g., for Transformers) operate on the flattened weight vector using the $L_2$ norm. Therefore, our use of the Frobenius norm is consistent with the standard SAM definition applied to matrix-valued weights.
>
> Random Perturbation. We use gradient estimation but not random direction: While the direction $U$ is random for a single step ($q=1$), it serves as a Randomized Gradient Estimator (RGE). As proven in Theorem 3.1 and Eq 3, the expectation of this estimator aligns with the true gradient, and not just a random direction. Our ablation in Table 8 confirms this: replacing our random ZO estimator with actual backpropagation yields <0.1% accuracy gain but increases training time by ~25%.
>
>
> **Key ablation studies.**
>
> (1) Placement: We agree with the reviewer’s suggestion. In the revision, we moved the key ablation studies on Zero-Order estimation and SVD strategies (were in Appendix D) into the main body to better highlight the contributions of the core components.
>
> (2) Isolation of Components & Performance Drivers: We appreciate the opportunity to clarify the contribution of the core algorithm versus the correction term. We respectfully note that our experimental design already includes a comprehensive ablation study across all experiments, explicitly reporting results for both "Lorenza" (the core algorithm) and "Lorenza+" (Core + Correction Term) side-by-side.
>
> Contrary to the impression of marginal gains, the core Lorenza algorithm (without the correction term) significantly outperforms state-of-the-art baselines on its own:
> Pre-training (Table 3): On LLaMA-1B, the core Lorenza algorithm achieves a perplexity of 14.93, significantly outperforming GaLore (15.64) and LoRA (19.21).
> Reasoning (Table 4): On the GSM8K benchmark (LLaMA-1B), the core Lorenza algorithm achieves 76.4% accuracy, surpassing GaLore (74.9%) and LoRA (68.3%).
> These results confirm that the core algorithm (Low-Rank SAM + ZO Estimator) is the primary driver of the SOTA performance and generalization capabilities. The "Lorenza+" variant provides an additional layer of improvement (e.g., boosting GSM8K to 77.1%), but the fundamental robustness and generalization gains are established by the base method. We will revise the text to explicitly interpret the "Lorenza vs. Lorenza+" columns as the isolation study for the correction term.

---

> ### Author Response · Authors · 2025-12-09
>
> **Minor changes**
>
> **GaLore Citation:** In the revised version, we corrected the statement to clarify that GaLore primarily reduces optimizer states via low-rank projection, rather than the number of trainable parameters (unlike LoRA).
>
> **Ablation Studies:** We agree with the reviewer's suggestion. We moved the key ablation studies (specifically the comparison of Correction Term vs. Low-Rank Base, and ZO vs. Backprop) from the Appendix to the main body to better isolate the source of our performance gains.
>
> **Hyperparameter:** All hyperparameters were mentioned in the appendix and have now been moved to the end of the experiment section.
> Reported SOTA Results in Table 3:  The exact setting associated with Table 3 was considered in the original papers of the SOTA methods we compare with. Thus, we report the numerical values presented in the relevant original SOTA papers and clearly state it in the revision.
>
> [1] Towards Understanding the Role of Sharpness-Aware Minimization Algorithms for Out-of-Distribution Generalization. Schapiro et al. 2024.
>
> [2] ASAM: Adaptive Sharpness-Aware Minimization for Scale-Invariant Learning of Deep Neural Networks. Jungmin et al 2021.
>
> [3] Global Sharpness-Aware Minimization Is Suboptimal in Domain Generalization: Towards Individual Sharpness-Aware Minimization. Song et al. ICLR 2025.
>
> [4] Towards Understanding Sharpness-Aware Minimization. Andriushchenko et al. ICML 2022

---

### Decision · Action_Editor_zsrT · 2026-02-02

**Recommendation:** Accept with minor revision

**Additional Comments:**

The reviewers were favorable, and by my reading, the paper also merits publication. In particular, I find the blending of these ideas of projections, zero order and sharpness aware minimization to be interesting, and the experiments to be well designed, and the results convincing. I do however would like to request some small changes, and remarks regarding the theorems and peak memory usage.

**Bounded loss assumption**:
In your Theorems, you use the very unusual assumption that the full batch loss is bounded over an unbounded domain. That is, that $max_W E_{\xi}[f(W,\xi)]$ is bounded.  This essentially hides that the parameter domain must be compact, which is also unusual. Specially when you consider that essentially all neural networks are coercive functions. By and large, this assumption is only really acceptable if you explicitly assume a bounded domain. This assumption makes it very hard to compare your results to those in the literature, and you need to be clear about this. In particular,  I checked your references such as  (Si & Yun, 2023) and Theorem 1 in (Sun et al., 2023)), and they use standard assumptions (Smoothness or Lipschitzness or bounded variance ..etc). So your theoretical results are not directly comparable to theirs. Can you please add a very explicit and careful remark on this assumption, and how it was not used in the related work. I only want clarity here. If you disagree with needing a remark, feel free to push back and disagree with me, but please explain your reasoning.


**Memory usage:** The paper needs a clear discussion on peak memory usage, which is often the real bottle neck for fine-tuning on a personal computer. Having to perform even one backwards pass, means there must be sufficient memory for the peak memory usage of the backward pass and/or optimizer states. Altering a method from using 2 backwards passes to 1 backward pass per iteration, might not affect the peak memory usage. Whereas, your use of a projected space for computing momentum buffers does directly decrease peak memory usage. Please add a remark on peak memory usage.

**Some minor comments:**

1. Top of page 4, looks like a lost sentence “  Another direction focuses on optimizing training methods,
reduces optimizer states memory consumption via low-rank gradient projection ,”

2. Middle of Page 6,

“Other variants of SAM (such as Theorem 3.5
in (Si & Yun, 2023) and Theorem 1 in (Sun et al., 2023)), Theorem 3.1 shows”

At the very least the above sentence is missing some connectors.

3. Page 6, it appears that $G_t^j$ is not defined in the main text. This doesn’t help my reading of Theorem 3.2, whose statement is a bit hard to parse. Please use additional space in your final revision, to make this statement more legible.

**Audience:**

Yes

**Audience Explanation:**

The paper introduces some interesting combinations of  zero-order gradient estimation, low-rank gradient updates, and Sharpness-Aware Minimization (SAM). It provides a good discussion, and new ideas, on decreasing memory usage, while still being able to train/fine tune models that attain good generalization.

**Claims And Evidence:**

Yes

**Claims Explanation:**

The paper is clear, and the claims do match the theoretical and experimental results.

---

> ### Author Response · Authors · 2026-03-04
> **Camera-ready version has been uploaded**
>
> Thanks for the helpful feedback. We have uploaded a camera-ready version that addresses the minor issues you raised and includes the requested clarifying remarks:
>
> Remark 3.3: Clarifies the bounded-loss assumption in Theorems 3.1–3.2, its technical role in our zeroth-order analysis, and its relation to assumptions used in prior work.
>
> \item Remark 3.4: Adds a clear discussion of memory, explicitly distinguishing optimizer-state vs.\ SAM overhead and addressing peak GPU memory (including the impact of avoiding two backward passes via BPFP).
>
> We also incorporated the remaining small editorial fixes you pointed out to improve readability.
>
> We hope these additions resolve the concerns and improve clarity.